# Stabilization of mineral-associated organic carbon in Pleistocene permafrost

Jannik Martens [1,2] ✉, Carsten W. Mueller [3,4], Prachi Joshi [5], Christoph Rosinger [6,7,8], Markus Maisch [5], Andreas Kappler [5,9], Michael Bonkowski [6], Georg Schwamborn [10,11], Lutz Schirrmeister [10] & Janet Rethemeyer [1] ✉

Ice-rich Pleistocene-age permafrost is particularly vulnerable to rapid thaw, which may quickly expose a large pool of sedimentary organic matter (OM) to microbial degradation and lead to emissions of climate-sensitive greenhouse gases. Protective physico-chemical mechanisms may, however, restrict microbial accessibility and reduce OM decomposition; mechanisms that may be influenced by changing environmental conditions during sediment deposition. Here we study different OM fractions in Siberian permafrost deposited during colder and warmer periods of the past 55,000 years. Among known stabilization mechanisms, the occlusion of OM in aggregates is of minor importance, while 33-74% of the organic carbon is associated with small, <6.3 μm mineral particles. Preservation of carbon in mineral-associated OM is enhanced by reactive iron minerals particularly during cold and dry climate, reflected by low microbial $CO_2$ production in incubation experiments. Warmer and wetter conditions reduce OM stabilization, shown by more decomposed mineral-associated OM and up to 30% higher $CO_2$ production. This shows that considering the stability and bioavailability of Pleistocene-age permafrost carbon is important for predicting future climate-carbon feedback.

Arctic permafrost stores about 1300 ± 200 Gt organic carbon (OC) of which about one third is present in the uppermost meter (472 Gt), while the larger part (834 Gt) is locked in frozen sediments below 3 m depth[1]. About 25–36% of this deep OC pool (329 to 466 Gt C[2]) is stored in silty to fine sandy deposits called Yedoma Ice Complex deposits (hereinafter referred to as Yedoma[2,3]) that were deposited in unglaciated areas of the circum-Arctic region during the late Pleistocene, with thicknesses of up to 40 m[1,3,4]. Because of their high ground ice content (with ice wedges and segregated ground ice of up to 50–80% of volume[3,4]), these sediments are particularly prone to rapid thaw upon climate warming in the Arctic region[3,5], which has already resulted in significant permafrost warming[6]. The melting of ground ice causes structural sediment collapse[7] and thus exposes ancient sedimentary organic matter (OM) to microbial degradation[8] that could increase emissions of carbon dioxide ($CO_2$) and methane ($CH_4$) to the atmosphere[9,10]. The magnitude of the greenhouse gas emissions may be up to 141 Gt of permafrost-bound OC by the end of this century[11] causing a positive feedback loop to climate change[12]

[1]Institute of Geology and Mineralogy, University of Cologne, Cologne, Germany. [2]Lamont-Doherty Earth Observatory, Columbia University, New York, NY, USA. [3]Chair for Soil Science, Technical University of Munich, Freising, Germany. [4]Department of Geosciences and Natural Resource Management, University of Copenhagen, Copenhagen, Denmark. [5]Department of Geosciences, University of Tübingen, Tübingen, Germany. [6]Institute of Zoology, University of Cologne, Cologne, Germany. [7]Institute of Agronomy, University of Natural Resources and Life Sciences, Tulln an der Donau, Austria. [8]Institute of Soil Research, University of Natural Resources and Life Sciences, Vienna, Austria. [9]Cluster of Excellence: EXC 2124: Controlling Microbes to Fight Infection, Tübingen, Germany. [10]Alfred-Wegener-Institute Helmholtz Centre for Polar and Marine Research, Permafrost Research Section, Potsdam, Germany. [11]Eurasia Institute of Earth Sciences, Istanbul Technical University Maslak, Istanbul, Turkey. ✉e-mail: jmartens@ldeo.columbia.edu; janet.rethemeyer@uni-koeln.de

and constituting one important 'tipping element' of the climate system[13].

The extent of the permafrost-climate feedback depends on the OC stock and the overall bioavailability and degradability of the previously freeze-locked OM, which is thought to be strongly related to its chemical composition and stage of degradation[14,15]. As the OM in Yedoma deposits originates from tundra-steppe vegetation[16,17] and accumulated at relatively fast rates[2,3,18], it is postulated to have experienced little decomposition and to contain large amounts of labile compounds[19–22] that are easily degradable[23,24]. This hypothesis is supported by high respiration rates measured in incubation experiments implying high bioavailability of the ancient OM upon thaw[25–27]. In contrast, other studies report lower $CO_2$ production rates for Yedoma OM compared to other permafrost OC pools[28], along with relatively low OC to total N ratios[2,16], suggesting this OM contains material from microbial decomposition that is therefore less bioavailable.

Besides chemical characteristics, physico-chemical protection mechanism may also affect degradation rates of the OM in Yedoma including the occlusion of particulate plant-derived OM (oPOM) within aggregated clusters of mineral particles, thus restricting microbial access to the so-called occluded particulate OM[14,29–31]. This mechanism significantly affects OC cycling in temperate soils[30,31], but its relevance in permafrost systems and particularly in Yedoma is unclear. Prior research on OM stabilization in permafrost-affected surface soils demonstrated the sequestration of OC as oPOM, however, with lower contribution to total OC storage compared to temperate soils[32–34]. Organic matter may also be stabilized by physico-chemical interactions[35] with fine-grained mineral particles that have high specific surface areas, allowing the formation of mineral-associated organic matter (MAOM)[36]. Previous work on the seasonally thawing active layer of permafrost soils showed that up to 50% of the bulk soil OC was stored in the fine-grained mineral soil fraction (<6.3 μm; from here on referred to as MAOM$_{<6.3μm}$). This fine silt and clay-sized MAOM$_{<6.3μm}$ consisted of substantially decomposed OM[37] with higher $^{14}C$ ages due to lower microbial turnover and lower bioavailability in incubation experiments than bulk OC[38], which underlines the possible preservation of OC over long time scales. The stabilization of OC as fine-grained (<6.3 μm) MAOM is promoted by sorption to reactive iron Fe(III) minerals (e.g., ferrihydrite or goethite)[35,39,40] and co-precipitation of Fe-OM[41,42]. However, thawing of permafrost also changes redox conditions due to waterlogged conditions, which may cause Fe(III) mineral reduction and dissolution causing the release of Fe-bound OC[43–46].

In this study, we investigate whether OM stored in Yedoma permafrost is protected against microbial degradation by occlusion within aggregates and/or formation of MAOM, and, more specifically, if these stabilization mechanisms trace back to the contrasting climatic conditions during sediment deposition in the late Pleistocene. Our study builds on two permafrost cores from Bol'shoy Lyakhovsky Island, NE Siberia (Fig. 1) located in the center of the Yedoma region[2,47]. Core L14-02 was recovered from a Yedoma hill that comprises undisturbed Yedoma sediments deposited during a cold period 33–55 ka before present (BP) for which widespread sediment and OM deposition is documented across large areas in Northeastern Siberia[2,16]. In addition, core L14-05 was recovered from a basin formed by thermokarst processes when the climate in Siberia was up to 4 °C warmer than today[48] around 15–11 ka BP. This core thus represents a sediment facies (in the following termed "thermokarst sediments") typical for the deglacial period during which thermokarst processes caused ground ice melting and ground subsidence[2]. We investigate the state of OM decomposition in bulk sediments from selected depth intervals representing different climatic conditions using elemental analysis (OC and N) and lipid biomarkers (n-alkanes). Possible OM stabilization by MAOM and aggregate formation is studied by analyzing the amount of OC in

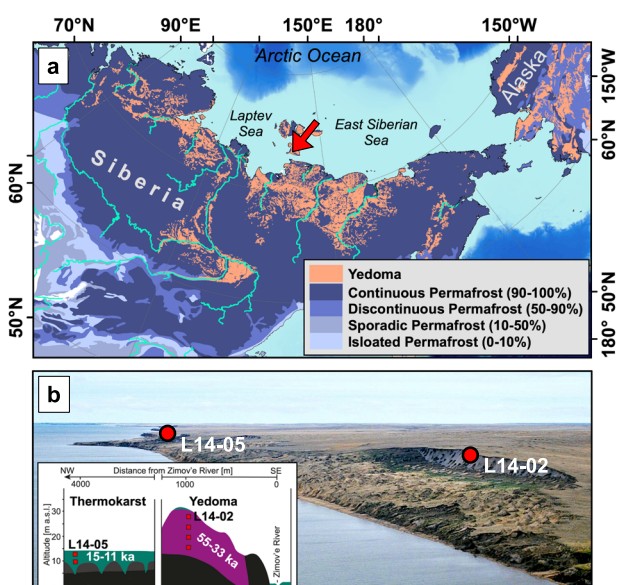

**Fig. 1 | Location of the study site in Arctic Siberia. a** includes an overview map with the study site indicated by the red arrow, the distribution of Yedoma deposits shown in orange color[69,70], and the different permafrost zones as blue shades[71]. **b** shows an aerial overview photograph of Bol'shoy Lyakhovsky Island where red markers indicate the locations of the permafrost drill cores L14-02 and L14-05. Also shown is the local stratigraphy with positions of the six samples of a Yedoma hill (L14-02) and a thermokarst basin (L14-05) marked in red. Data for bathymetric base map of panel a based on IBCAO[72]; photograph in panel **b** by G. Schwamborn was previously published by Zimmermann, H. et al. The History of Tree and Shrub Taxa on Bol'shoy Lyakhovsky Island (New Siberian Archipelago) since the Last Interglacial Uncovered by Sedimentary Ancient DNA and Pollen Data. Genes (Basel) 8, 273 (2017).

density and particle-size fractions, its $^{14}C$ contents, and the stage of OM degradation by $^{13}C$ solid-phase nuclear magnetic resonance spectroscopy ($^{13}C$-NMR). In addition, reactive Fe in MAOM$_{<6.3μm}$ is analyzed by $^{57}Fe$ Mössbauer spectroscopy in order to determine the different Fe mineral phases promoting OC stabilization. Potential changes in the mineral composition of the sediments were analyzed using X-ray diffractometry (XRD). The bioavailability of MAOM$_{<6.3μm}$ is investigated by measuring microbial basal respiration rates. Since the microbial biomass was killed by the application of sodium polytungstate during sediment fractionation, we obtained a microbial inoculum from untreated permafrost samples and added this to the <6.3 μm fraction prior to incubation for 108 h at 21 °C. Only the $CO_2$ production after this initial microbial growth phase and depletion of added OM (i.e., basal respiration) is considered in this study[49].

## Results and discussion

### Compositional characteristics of OM in bulk sediment

The climatic conditions during Yedoma deposition have varied considerably over the 44 ka long depositional period resulting in differences in vegetation[48,50] and microbial decomposition of the sedimentary OM[24]. We used the OC/N ratio and the carbon preference index (CPI) of n-alkanes ($C_{21}$–$C_{35}$) as indicators for the stage of OM degradation and alteration, respectively. During OM decomposition, OC is removed from the sediment while N is retained resulting in low OC/N ratios[15]. Similarly, low CPI values indicate the degradation of typical long-chain plant-wax lipids[51,52]. The oldest Yedoma material investigated here was deposited when climatic conditions were cold and dry, allowing only a sparse tundra-steppe vegetation cover during 55–48 ka BP[17,48]. Accordingly, these sediments have lower OC content (13 and 14 mg OC g$^{-1}$) due to low productivity compared to deposits from the subsequent moderately

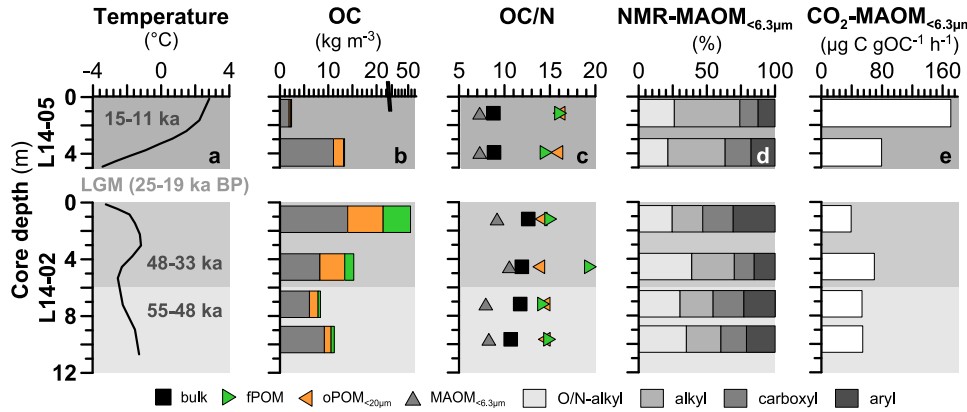

**Fig. 2 | Mass partitioning of bulk organic matter (OM), composition of OM fractions and the bioavailably of mineral-associated OM (MAOM$_{<6.3\mu m}$).** Data is shown for thermokarst sediments (L14-05) and Yedoma deposits (L14-02) under changing climate during the late Pleistocene, along with **a** Siberian paleo-temperatures (based on ref. [48]) for the three depositial time frames (55–48, 48–33, and 15–11 ka before present–BP); No material was deposited during the Last Glacial Maximum (LGM 25–19 ka BP); Further shown are **b** bulk organic carbon (OC) of mass fractions in kg m$^{-3}$; **c** OC/N ratios of all fractions in colored symbols (green for free particulate OM–fPOM; orange for occluded particulate OM–oPOM; dark gray for MAOM$_{<6.3\mu m}$) and the bulk OC as black symbols; the composition of MAOM$_{<6.3\mu m}$ based on $^{13}$C Nuclear magnetic resonance (NMR) is shown in gray shades in panel **d**; while panel **e** shows the microbial mineralization flux of MAOM$_{<6.3\mu m}$ to $CO_2$.

warm interval between 48–33 ka BP[17,48] (26 and 47 mg OC g$^{-1}$; Fig. 2; Supplementary Table 1). The slightly warmer climatic conditions promoted vegetation growth in this tundra environment and produced higher water contents in the sediments, which enhanced production and accumulation of fresh OM[16,17,24]. This is reflected by higher CPI values (6.4 and 6.9) of the OM when compared to the earlier, colder period (55–48 ka BP; CPI: 1.9 and 4.8; Supplementary Table 1), indicating less degraded OM was sequestered during this period. The warmer period was followed by very cold and dry conditions during the last glacial maximum (until ca. 19 ka BP[48]), during which no sediment was accumulated or it was eroded at the study site. With temperatures of up to 4 °C warmer than today, the climate during the post-glacial transition 15–11 ka BP[48] caused tremendous transformation of Arctic permafrost deposits at a large scale[53,54]. The rapid temperature rise during this period promoted plant growth[48,50], while thermokarst processes promoted the degradation of the sedimentary OM[55,56]. As a result, the thermokarst sediments from this period have lower OC contents (4 and 15 mg C g$^{-1}$), OC/N ratios (8.8 vs. 10.8–12.7), and CPI values (4.8 and 5.0; Supplementary Table 1) compared to the Yedoma deposited during both earlier periods (55–48 and 48–33 ka BP). These data confirm a significant influence of climatic conditions and associated change in vegetation and microbial activity to the state of OM degradation in late Pleistocene permafrost deposits[16,24].

## Stabilization and bioavailability of MAOM

Among the OM fractions recovered from the two permafrost cores, we found most of the OC stored as MAOM$_{<6.3\mu m}$ (33–74% of total OC; Fig. 2; Supplementary Table 2). This range agrees well with data published for several permafrost soils (including the seasonally thawed active layer) from around the circum-Arctic[32,33,38] and other Yedoma deposits from Northeastern Siberia[26], suggesting that MAOM$_{<6.3\mu m}$ holds a large portion of the OC stored in Yedoma permafrost systems. However, permafrost sediments of the three depositional time frames investigated in the present study (55–48, 48–33, and 15–11 ka BP) exhibit large differences in the amount and composition of MAOM$_{<6.3\mu m}$, which likely correspond to the contrasting climatic conditions and resulting changes in landscape conditions during the late Pleistocene.

The cold and dry glacial climate during the Yedoma deposition around 55–48 ka BP promoted MAOM formation, reflected by the storage of more than half of the total OC as MAOM$_{<6.3\mu m}$ (54 and 73%;

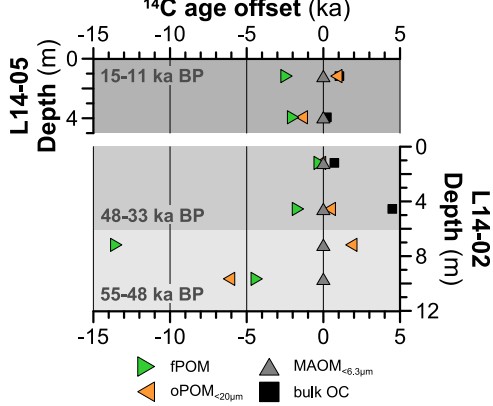

**Fig. 3 | Relative radiocarbon ($^{14}$C) age offsets (in 1000 years-ka) between organic matter (OM) fractions and bulk organic carbon (OC).** The different OM fractions are shown as colored triangles (green for free particulate OM–fPOM; orange for occluded particulate OM–oPOM; dark gray for mineral-associated OM-MAOM$_{<6.3\mu m}$) and bulk OC is shown as filled black square symbol. All results are normalized to the ages of the MAOM$_{<6.3\mu m}$ fraction. The bulk OC of the two deep samples in L14-02 revealed infinite $^{14}$C ages and were excluded from this figure.

6.1 and 9.2 kg m$^{-3}$; Fig. 2). The prevalent environmental conditions also resulted in the largest amounts of poorly crystalline Fe(III) (oxyhydr) oxides (based on Mössbauer spectra collected at 77 K and 5 K) in MAOM$_{<6.3\mu m}$ (35 and 37%) that enhance OM stabilization[35,39] (Supplementary Table 6). Low inputs of plant litter and strong association of OM with mineral particles are mirrored by relatively low $CO_2$ production rates during the basal respiration measurements (53 ± 8 and 54 ± 7 µg $CO_2$-C per gOC per h; Fig. 2, Supplementary Table 7). Low turnover rates due to high stabilization of OC are further shown by the highest age offsets between MAOM$_{<6.3\mu m}$ and fresh plant material present as free POM (fPOM) (4.3 and 13.5 ka; Fig. 3; Supplementary Table 3). As a result, MAOM$_{<6.3\mu m}$ contains relatively high amounts of degradation-resilient compounds such as alkyl C (24 and 25%; e.g., lipids, suberin, or cutin[57]), aryl C (21 and 22%; aromatic compounds[58]) and carbonyl/carboxyl/amide C (19 and 22% from fatty acids, fatty ester[58]) identified by $^{13}$C-NMR spectroscopy (Fig. 2), while only 30 and 35% constitute more labile O/N-alkyl C compounds (e.g., carbohydrates[57]).

Total sedimentary OC accumulation was two to three times higher during slightly warmer and wetter conditions between 48 and 33 ka BP than in the preceding colder period, due to an enhanced vegetation growth, input of fresh OM and, reduced decomposition under higher soil moisture. This resulted in notably larger amounts of OC in the form of fPOM and oPOM$_{<6.3\mu m}$ (Fig. 2) and in similar, to larger amounts of OC in MAOM$_{<6.3\mu m}$, when compared to the preceding colder period (55–48 ka BP), with 8.2 kg m$^{-3}$ (36% of the bulk OC) at 48 ka BP and 14 kg m$^{-3}$ (33%) at 33 ka BP. The very small $^{14}$C age offsets between MAOM$_{<6.3\mu m}$ and fPOM (0.2 to 1.7 ka; Fig. 3) indicate low stabilization of OC during the 48–33 ka period, which is also reflected by higher CO$_2$ production during incubations of MAOM$_{<6.3\mu m}$ from around 48 ka BP (70 ± 13 µg CO$_2$-C per gOC per h), although less clearly by MAOM$_{<6.3\mu m}$ from the later phase (33 ka BP) of this moderately warm period (39 ± 3 µg CO$_2$-C per gOC per h; Fig. 2). The lower stabilization may be related to a decreased amount of Fe(III) (oxyhydr)oxides in MAOM$_{<6.3\mu m}$ (20 and 27%) under less oxic conditions in a generally more water saturated active layer during this depositional period[17]. Moreover, the highest OC/Fe mass ratio of 1.1 at 33 ka BP—which is notably higher than the sorption capacity of Fe(III) oxides (0.22)[59,60]—may show that co-precipitation of Fe-MAOM, in addition to mineral absorption, was more important under such varying oxic conditions[61] than during the previous colder and dryer climate. Further, the MAOM$_{<6.3\mu m}$ seems to consist of less bioavailable OC, shown by large amounts of degradation-resilient compounds including 31% alkyl (2.6 kg m$^{-3}$), up to 30% aryl-C (4.2 kg m$^{-3}$) and slightly higher alkyl/O/N-alkyl ratios (0.8 and 0.9). Despite lower stabilization and bioavailability of the MAOM$_{<6.3\mu m}$, we note that the total CO$_2$-C production potential in the 48–33 ka old sediments is 36% larger (0.55 ± 0.05 and 0.57 ± 0.11 mg CO$_2$-C per m$^3$ per h) as more MAOM$_{<6.3\mu m}$ is available per m$^3$ when compared to the cooler period (0.33 ± 0.05 and 0.50 ± 0.06 mg CO$_2$-C per m$^3$ per h; 55–48 ka BP), which outweighs the differences among the MAOM$_{<6.3\mu m}$ fractions. It thus appears that the warmer climate has led to an overall higher bioavailability of OM and lower OC stabilization by MAOM.

During the deglacial period, the amount of OC in MAOM$_{<6.3\mu m}$ decreased from highest values of 11 kg m$^{-3}$ (74% of total OC) around 15 ka BP to lowest values of 1.9 kg m$^{-3}$ in the youngest sediment investigated from around 11 ka BP (45% of total OC; Fig. 2). This change is most probably to be related to enhanced microbial degradation of the OM due this time of rising temperatures, which is reflected by up to three times higher CO$_2$ production rates of MAOM$_{<6.3\mu m}$ rising from 79 ± 11 to 171 ± 11 µg CO$_2$-C per gOC per h in this period (Fig. 2; Supplementary Table 7) and relatively low $^{14}$C age offsets between MAOM$_{<6.3\mu m}$ and fPOM (Fig. 3). The wetter, microoxic or even anoxic conditions may be an additional factor leading to lower OC stabilization and higher bioavailability of OM due to reductive dissolution of Fe-OM structures[43,61], particularly in thermokarst basins[17,48], as shown by a slightly lower content of Fe(III) (oxyhydr)oxides (22 and 28%; measured by Mössbauer spectroscopy at 77 K and 5 K) and less overall Fe(III) minerals (69 and 70% measured at 77 K in 15–11 ka BP vs. 73 and 76% in 55–33 ka BP Yedoma). As a result of low physico-chemical stabilization, MAOM$_{<6.3\mu m}$ is more strongly transformed with highest alkyl C values (41 and 47%; Fig. 4) and lowest O/N-alkyl C, i.e., carbohydrates (21 and 26%[33,62,63]), where the alkyl/O/N-alkyl degradation ratios are the highest among all samples (Supplementary Table 4). Low stability and high bioavailability of MAOM$_{<6.3\mu m}$ in 15–11 ka BP permafrost underscores that thermokarst processes may be an important driver for (future) CO$_2$ production.

## Occluded particulate OM

Low OM bioavailability due to the occlusion in aggregates as oPOM$_{<20\mu m}$ appears to be less important compared to the persistence of OC stored as MAOM$_{<6.3\mu m}$, as only a minor amount of about 8 to 23% of the total OC (equivalent to 0.3 to 7.3 kg m$^{-3}$) was present as

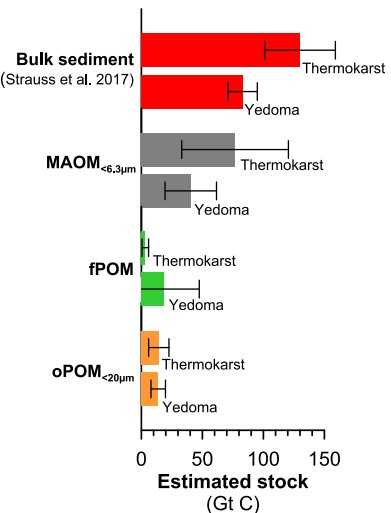

**Fig. 4 | Estimated carbon stock in bulk sediments and organic matter (OM) fractions.** The different organic carbon (OC) pools are mineral-associated OM (MAOM$_{<6.3\mu m}$ in gray), free particulate OM (fPOM in green) and occluded particulate OM (oPOM$_{<20\mu m}$ in orange) in Yedoma deposits and thermokarst sediments. The OC pools and uncertainties (with the standard deviation shown as error bars) were estimated based on published OC stocks (and its uncertainty estimates based on observational OC variability) in late Pleistocene permafrost[2] (shown as red bars), which is further described in the Methods.

oPOM$_{<20\mu m}$ in all samples. Nonetheless, OC in the oPOM$_{<20\mu m}$ fractions had very similar residence times ($^{14}$C-OC ages) like MAOM$_{<6.3\mu m}$ (Fig. 3) in most time intervals and thus also seems to be less accessible for microorganisms although it was more labile than OC in MAOM$_{<6.3\mu m}$. The OC stored in oPOM$_{<20\mu m}$ from Yedoma of the coldest period at 55–48 ka BP (11 and 16% of the total OC; (1.4 and 1.8 kg m$^{-3}$), consisted predominantly of labile O/N-alkyl C (41 and 52%; Fig. 4) from carbohydrates[57]. Warmer climatic conditions during 48–33 ka BP increased the portion of easily degradable[64] O/N-rich compounds[57] (41 to 61%; Supplementary Table 4), similar to that of fPOM (46 and 57%), while oPOM$_{<20\mu m}$ fractions were also slightly larger during this interval (17 and 23%, 7.3 and 5.1 kg m$^{-3}$). Deglacial warming resulted in lower amounts of OC preserved as oPOM$_{<20\mu m}$ (8.2 and 14% of total OC; 0.3 and 2.1 kg m$^{-3}$), with losses of labile organic compounds like O/N-alkyl C (38 and 39%) in the oPOM$_{<20\mu m}$ representing OM degradation during this warm climate when compared to the preceding colder periods. Degradation and decline of oPOM$_{<20\mu m}$-associated OC might indicate that aggregate occlusion is less important for OC stabilization among the mineral permafrost sediments studied here. However, even if the oPOM fraction stores little OC in the Yedoma, it may play a significant role in OM stabilization in more OM-rich peat-like permafrost deposits[32,34].

## Implications for large-scale carbon cycling

We showed that substantial amounts of OC in late Pleistocene permafrost are stabilized by organo-mineral interactions. While large proportions of total OC were associated with fine silt and clay-sized minerals (i.e., MAOM$_{<6.3\mu m}$; 33–74%), much less of the total OC (8.2–23%) was stored as occluded particulate OM in aggregated structures (e.g., oPOM$_{<20\mu m}$). The OC stocks and the bioavailability of MAOM$_{<6.3\mu m}$ were significantly affected by the varying climatic conditions during Yedoma deposition between 55 and 11 ka BP. The cold and dry conditions between 55–48 ka BP reduced plant-litter inputs and the state of OM decomposition was low. During this period 54-73% of the total OC were efficiently stabilized as MAOM$_{<6.3\mu m}$, especially due to contributions of poorly crystalline Fe (oxyhydr)oxides. The persistence of this OC pool was supported by low microbial CO$_2$

production during the respiration experiments. Warmer conditions around 48–33 ka BP promoted vegetation growth, resulting in about seven times larger amounts of more labile particulate OM and about 50% more OC in $MAOM_{<6.3\mu m}$. However, the lower stage of OM degradation and reduced MAOM stability due to higher soil moisture and varying oxic conditions enhanced OM bioavailability and contributed to a 36% higher potential for $CO_2$ production. The warm and wet deglacial climate induced thermokarst processes (the thaw and collapse of permafrost and subsidence of land surface), further reducing OC stabilization of the $MAOM_{<6.3\mu m}$ pool between 15 to 11 ka BP. Higher moisture and presumably waterlogged and anoxic conditions decreased MAOM formation, or even caused reductive dissolution of Fe(III) (oxyhydr)oxides, which led to less stable OC stored as MAOM and the highest $CO_2$ production in respiration experiments.

Our work suggests that OC sequestration as $MAOM_{<6.3\mu m}$ was inherently involved in the accumulation of OC in permafrost sediments throughout different climates during the late Pleistocene. We therefore hypothesize that MAOM formation in Yedoma occurred concurrently at other sites across Northeastern Siberia and Alaska at the time, likely representing a significant OC pool in the Arctic Yedoma region. Based on published stock estimates of OC in Yedoma sediments[2] and mass contributions of $MAOM_{<6.3\mu m}$ at our study site, we estimate that about $40 \pm 21$ Gt of the OC that is freeze-locked in Yedoma may be present as $MAOM_{<6.3\mu m}$ (Fig. 4; Supplementary Table 8). By contrast, post-glacial thermokarst sediments may hold $77 \pm 44$ Gt OC as $MAOM_{<6.3\mu m}$. We like to emphasize that these numbers build on a limited observational dataset and constitute first-order estimates that are attributed to uncertainty. While future research will improve accuracy of such estimates, the present study suggests that a total of about $117 \pm 65$ Gt OC in ice-rich permafrost deposits may be stored as MAOM, which adds complexity to the bioavailability of the sequestered OC. Contrasting stability and bioavailability of Pleistocene-age OC deposited during different climate periods should be considered when anticipating future permafrost thaw and the potential for greenhouse gas production and climate-carbon feedback.

## Methods
### Study sites and sampling
The permafrost cores investigated in this study were recovered from Bol'shoy Lyakhovsky Island, the southernmost island of the New Siberian Islands Archipelago between the Laptev and East Siberian seas (Fig. 1). The cores were drilled using a mobile drilling rig (KMB3-15M) during a joint German-Russian expedition in April 2014[24,47,50]. An age model based on AMS $^{14}$C data of plant remains was published elsewhere[50].

Core L14-02 (73.33616° N; 141.32776° E) was taken on a typical Yedoma hill 1.2 km northwest of the Zimov'e River mouth. The upper 11.26 m consists of silty-sandy sediments with macroscopic organic remains and an alternation of horizontal, vertical, and reticulated ice veins and lens-like cryostructures[24] deposited under sub-aerial conditions ca. 55–33 ka BP[17,24,50]. Four subsamples were taken from selected depth intervals deposited between 48–33 ka BP (1.2 and 4.6 m core depth) and during 55–48 ka BP (7.2 and 9.7 m core depth). Below 11.26 m the core was drilled in an ice wedge from which no samples were taken.

Core L14-05 (73.34994° N; 141.24156° E) was drilled 4.3 km northwest of the Zimov'e River mouth in a thermokarst depression formed during the post-glacial transition -15–11 ka BP[65]. The 7.89 m long core is composed of lacustrine, boggy sediments, and sub-aerial sediments deposited after lake drainage during the last deglaciation[48,50]. We took two subsamples from the lower lacustrine (3.9 m core depth) and the upper sub-aerial (1.2 m core depth) part to represent thermokarst sediments for this study.

The core segments were kept frozen until subsampling in a climate chamber (−10 °C) using a band saw.

### Sediment fractionation
We performed a combined density and particle-size fractionation to separate the sedimentary OM into fractions representing different stabilization mechanisms. These included free particulate organic matter (fPOM), occluded particulate organic matter (oPOM), as well as grain size classes that included coarse and medium sand (>200 μm), fine sand (>63 μm), coarse silt (>20 μm), medium silt (>6.3 μm), and fine silt and clay (<6.3 μm)[66]. Briefly, 40 g of oven-dried sediment was suspended in a solution of sodium polytungstate (1.8 g cm$^{-3}$) and allowed to settle overnight. Floating material (<1.8 g cm$^{-3}$) represents fPOM, which was washed several times with MilliQ-water and dried. The remaining sediment (>1.8 g cm$^{-3}$) was sonicated (440 J ml$^{-1}$) to break up potential aggregates and release oPOM. The suspension was separated from the sediment and centrifuged to collect oPOM, which was repeatedly filtered through 20 μm mesh size under using a vacuum pump until the suspensions electric conductivity dropped below 5 μS cm$^{-1}$. Resulting oPOM particles smaller than 20 μm (oPOM$_{<20\mu m}$) were dried and heavier sediment residues were further separated using wet sieving into sand (>63 μm) and coarse silt (>20 μm). The smaller particles sizes were separated using sedimentation to recover the <6.3 μm fraction, which contains $MAOM_{<6.3\mu m}$. All fractions were freeze-dried, weighed (Supplementary Table 2) and further analyzed. The recovery of sediment mass after fractionation was between 96–98% of the initial sample weight.

### Elemental analysis
Total OC and N contents of OM fractions and of bulk samples were measured using a Vario MICRO cube (Elementar Analysensysteme GmbH, Germany). Inorganic carbon was removed from the sediment prior to analysis with HCl (1% at 60 °C overnight). Based on repeatedly run standard materials of different OC and N concentrations, the 1-sigma measurement uncertainty of the combined measurement is 1% for OC and 3% for N and resulting OC/N ratios.

### Mass partitioning of organic carbon and losses
To assess the distribution and stock of OC in OM fractions we multiplied the fraction yields (wt% of the bulk sediment) with the OC content of the fractions (%OC of the fraction). The three major OM fractions ($MAOM_{<6.3\mu m}$, fPOM, oPOM$_{<20\mu m}$) recovered between 64–100% of the OC determined for the bulk sediment. The density of OC per m$^3$ was calculated using the published average density of Yedoma (0.87 g cm$^{-3}$) and thermokarst sediments[4] (0.94 g cm$^{-3}$).

### Radiocarbon analysis of organic matter fractions
For radiocarbon analysis, inorganic carbon was removed from the bulk sediment and OM fractions (fPOM, oPOM$_{<20\mu m}$ and $MAOM_{<6.3\mu m}$) as described above. The dried sediments were then combusted in tin boats and converted to elemental C using an automated graphitization system (AGE). Radiocarbon analyses were performed with the 6 MV Tandetron Accelerator Mass Spectrometer (AMS; HVE, The Netherlands) at the AMS facility of the University of Cologne (Germany)[67]. The results are reported as conventional $^{14}$C ages and $\Delta^{14}$C in Supplementary Table 3 (both corrected for exogenous carbon contributions and isotopic fractionation), including a 1-sigma measurement uncertainty.

### NMR spectroscopy
Cross-polarization magic angle spinning nuclear magnetic resonance spectroscopy of $^{13}$C (CPMAS NMR, Bruker DSX 200, Bruker BioSpin GmbH, Karlsruhe, Germany) was used to identify major organic compound classes for a representative number of OM fractions. In order to minimize chemical anisotropy, samples were filled into zirconium dioxide and rotated at speed of 6.8 kHz in a magic angle whereas Hartmann-Hahn mismatches were avoided using a ramped $^1$H pulse during a contact time of 1 ms. For integration, chemical shift regions were used as given: alkyl C (−10–45 ppm), O/N-alkyl C (45–110 ppm),

aryl/olefine C (110–160 ppm), and carbonyl/carboxyl/amide C (160–220 ppm). The peak regions are reported in % of the total (Supplementary Table 4).

## Lipid biomarker analysis

Plant leaf wax-derived lipids including long-chain *n*-alkanes were extracted according to a protocol established at the University of Cologne[33]. Briefly, lipids were extracted from 5 g freeze-dried sediment using accelerated solvent extraction (ASE 350, Thermo Scientific, USA) and dichloromethane (DCM) and methanol (MeO; v-v) 9:1 and 1:1 20 min each at 100 bar and 120 °C. The resulting total extractable lipids were evaporated until dry under $N_2$. Bound lipids were release by saponification using a mixture of MeOH: KOH (95:5) for 2 h at 80 °C. The neutral lipid fraction was extracted after addition of MilliQ water with hexane and the *n*-alkanes were recovered by elution over $SiO_2$ columns using hexane[33]. The *n*-alkanes were analyzed using an Agilent 7890B (Agilent Technologies, USA) gas chromatograph (GC) equipped with a flame ionization detector (FID) and a DB-5MS column quantified using authentic external standards[33]. The 1-sigma measurement uncertainty of the measurements was 3%, based on a repeatedly analyzed *n*-alkane standard.

The stage of OM degradation and alteration, respectively, was assessed using the carbon preference index (CPI) of *n*-alkanes ($C_{21}$–$C_{35}$), which determines the odd over even predominance of the carbon chains, which decreases with progressing degradation resulting in lower CPI values[51]. Total concentrations ($C_{27}$, $C_{29}$, $C_{31}$) and CPI values of *n*-alkanes are given in Supplementary Table 1 and concentrations of every homolog in Supplementary Table 5.

## Stock estimates for carbon in organic matter fractions

We also estimated the pool size of $MAOM_{<6.3\mu m}$ and particulate OM ($fPOM$; $oPOM_{<20\mu m}$) for the Arctic region. To do so we extrapolated the amount of total OC in OM fractions in this study over published stock estimates[2] for deposits of the Yedoma domain in Siberia and Alaska, including Yedoma and deposits that accumulated after Yedoma degradation in thermokarst landform (thermokarst). The stock estimates are shown in Supplementary Table 8.

## Identification of minerals and iron mineral phases

For mineral analysis of bulk samples, mortared sediment samples were loaded onto a silica wafer and analyzed in a X-ray 2D-Diffractometer (Bruker D8 Discover with GADDS, μ-XRD2, Bruker AXS GmbH, Karlsruhe, Germany), using a cobalt anode tube as x-ray source with a Co-Kα wavelength of 1.541874 Å and a 2D detector with 40° angle cover (Bruker Våntec 500 Bruker AXS GmbH, Karlsruhe, Germany). Reflection pattern analysis and mineral identification were carried out using the Match! software for phase identification from powder diffraction (Match!, Crystal Impact, Bonn, Germany). Resulting XRD spectra are shown in Supplementary Fig. 1.

Iron mineral phases were identified using $^{57}Fe$ Mössbauer spectroscopy. Briefly, dried $MAOM_{<6.3\mu m}$ fractions were loaded into plexiglas holders (area 1 cm$^2$), forming a thin disc. Samples were kept in airtight jars at −20 °C until measurement. Holders were inserted into a closed-cycle exchange gas cryostat (Janis cryogenics) under a backflow of He to minimize exposure to air. Spectra were collected at 77 K and 5 K using a constant acceleration drive system (WissEL) in transmission mode with a $^{57}Co/Rh$ source. All spectra were calibrated against a 7 μm thick α-$^{57}Fe$ foil that was measured at room temperature. Analysis was carried out using the Recoil (University of Ottawa) and the extended Voigt Based Fitting (VBF) routine[68]. The half width at half maximum (HWHM) was constrained to 0.123 mm/s during fitting. The resulting Fe mineral phases are reported as % of the peak area and are shown in Supplementary Table 6, while the Mössbauer spectra at 5 K and 77 K are shown in Supplementary Figs. 2 and 3, respectively.

## Basal respiration measurements

For the inoculation of the <6.3 μm fractions, a soil slurry was prepared under sterile conditions the day prior to inoculation. Briefly, ca. 6.5 g of frozen, untreated permafrost soil from the corresponding region were dissolved in 50 ml of deionized water for 60 min under constant stirring on a magnetic stirrer. Subsequently, the soil slurry was filtered through a folded filter paper (Whatman 595) and stored in a sterile tube at 4 °C. The soil slurry was measured for dissolved organic carbon and total dissolved nitrogen concentrations.

For the respiration measurement, 1.5 g of dry <6.3 μm fractions were weighed into sterile glass vessels and 2.25 ml of the soil slurry was added, resulting in a water content of 60% (high amounts of soil slurry were necessary to fully moisten the fractions). The vessels were then immediately mounted on an automated electrolytic micro-respiratory apparatus[49]. The $O_2$ consumption rates were measured hourly at 21 °C for a total of 108 h and subsequently converted to $CO_2$ production using gas constants. The basal respiration represents the level of $CO_2$ production during the time interval after the initial microbial growth phase and the concurrent depletion of any OM that was added with the soil slurry (corresponding to a time interval between ca. 40 and 114 h). Results of the basal respiration measurements are shown in Fig. 2e, Supplementary Table 7 and expressed as mean ± 1-sigma uncertainty μg $CO_2$-C g$^{-1}$ per dry fraction per h and in μg $CO_2$-C g$^{-1}$ OC per h.

## Data availability

All data used and generated in this study are provided in the Supplementary Information. The data used in this study are also available under accession code https://doi.org/10.5281/zenodo.7644532.

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

## Acknowledgements

This study was financially supported by the German Ministry of Science and Education (BMBF) within the joint Russian-German project 'Carbo-Perm' (grant number 03G0836E). We thank all colleagues involved in the field campaigns to Bol'shoy Lyakhovsky Island in 2014 for drilling of sediment cores. Sandra Jivcov, Svetlana John, Dorothea Klinghardt, Ulrike Patt, Bianca Stapper, Ilona Steffen, Irene Brockhaus and Daniela Warok are thanked for their assistance and valuable advice during sample preparation at the University of Cologne. We also thank Stefan Heinze and Alfred Dewald for the AMS analyses of the bulk sediment and the OM fraction. AK acknowledges infrastructural support by the Deutsche Forschungsgemeinschaft (DFG, German Research Foundation) under Germany's Excellence Strategy, cluster of Excellence EXC2124, project ID 390838134.

## Author contributions

The research program and the field campaigns to Bol'shoy Lyakhovsky were developed and coordinated by J.R., L.S., and G.S. L.S. and G.S. were involved in the drilling and subsequent basic laboratory work. J.R. and C.W.M. coordinated the laboratory work and the writing of the paper. P.J., M.M., and A.K. contributed with mineral analysis, while C.R. and M.B. carried out the respiration measurements and data interpretation. J.M. executed the research, analyzed the samples, performed calculations and data analysis, drafted and coordinated the manuscript.

## Funding

## Competing interests

The authors declare no competing interests.
