## [Peer Review File · Nature Communications]

Stabilization of mineral-associated organic carbon in
Pleistocene permafrostReviewers' comments:

Reviewer #1 (Remarks to the Author):

This manuscript aims to determine the relative importance of mineral associated organic matter in yedoma deposits by analyzing two sediment cores from Siberia, one dated to Marine Isotope Stage 3 (cold & glacial) and one dated to the MIS 2 to 1 transition (warm). The authors analyzed bulk OC, ^{14}C ages (bulk and compound specific) and a suite of n-alkanes on bulk sediment and density/size fractionated sediment to compare OM pools in their two cores.

Yedoma carbon is a Pleistocene-era permafrost OC store that is hypothesized to be relatively labile and a potential source of large amounts of greenhouse gases as Arctic permafrost warms and thaws. The guiding hypothesis for this work is that if a significant amount of this yedoma carbon is mineral-bound, it will actually be less labile than assumed from previous chemical analyses and therefore potentially more stable and less of a source of greenhouse gases in a warming climate.

In general, the authors conclude that this is true: that much of the yedoma carbon, especially the younger yedoma (from MIS 2 -1 transition) is mineral bound and therefore less bioavailable than previously estimated. However, some of the estimates that they cite for the bioavailability of yedoma carbon are not based only on chemical studies but instead on microcosm studies, where the kinetics of degradation upon thaw were directly measured. While it is true that mineral bound OM is in general more protected from degradation than free OM, we have direct measurements of the degradation potential of yedoma carbon. It is also true that these direct measurements are from samples that are different from the ones analyzed here, and potentially not directly comparable, but this point is not addressed in the manuscript.

Further, the authors give relatively short descriptions with only a shallow treatment of their geochemical data. How does this chemical analysis compare directly with other analyses of the OM chemistry of yedoma carbon? What does this tell us about the chemistries of this OM that is spread out geographically and was deposited over large amounts of time? While the authors do a good job of comparing the OM chemistry of their three time points (early MIS 3, late MIS 3, and MIS2-1 transition) there is very little discussion of how this compares with earlier yedoma carbon studies.

In general, I think the findings in this manuscript are interesting and impactful, I think this study would benefit from a longer-form manuscript, with a fuller treatment of the data therein.

Reviewer #2 (Remarks to the Author):

Review of Martens et al

General comments:

This manuscript presents some good new data on the MAOM fraction of the Yedoma sediments and also provides some organic C characterization of those fractions. Aside from not presenting errors on most of the measurements—something that is desperately needed to understand the relationships between the samples and also a key requirement of NCOMM if I remember correctly—the work appears to be carried out in a highly skilled and careful, rigorous manner. Clearly this data is needed and will help move the work on permafrost C release upon melting ahead. However, as currently presented, this work suffers from some pretty serious interpretive missteps. I have highlighted many of these in the line comments below, and I summarize them briefly here at the top.

1. The authors are not actually measuring the bioavailability of the MAOM. This is inferred based on the size fraction of the OC, and somewhat by the observation that it appears to be already rather degraded based on C/N, ^{13}C NMR, and other C analyses. However, those measures apply to

conditions of formation, not the conditions of melting permafrost. To address this the authors look at the MIS 2/1 section of the sample and suggest at one point that this might be what post-thawed Yedoma deposits would look like. As I outline below, this exercise, to me, makes some interpretive errors by focusing on the fraction of total C as MAOM and the net abundance (concentration) of MAOM. I think the latter is the more relevant way to assess the data, and should at a minimum be discussed even if the authors disagree with me. But see below on this and back to the bioavailability point. I certainly agree that MAOM is likely to be of lower bioavailability, but the main issue here is that is not what is measured and the authors are misleading the reader by discussing bioavailability, especially in the abstract and concluding paragraph. Simply discuss this as the amount of C in the fine fraction, which the authors then define as MAOM. Be precise about what is being reported. If the authors had actually measured the bioavailability using an incubation or something, this would really strengthen the paper. If that is possible, I encourage them to do so.

2. We do not know the mineral composition of the minerals in the MAOM. Does this matter? Well, if they are strongly Fe-based minerals or comprised of layered silicate clays bound by Fe minerals, then they may be susceptible to release if the material is subjected to Fe reduction in an anoxic melt water, which is a common characteristic of these systems. Further this matters if the mineral composition changes across the different deposits as that may alter the amount of C that can be stored on mineral surfaces. Finally, it matters if the authors are interested in understanding how much of the MAOM will turnover should it become melted as the mineral composition becomes critical in this regard.

3. The authors have not discussed or assessed the if the C in the fine fraction near its approximate MAOM saturation level? [see e.g., Stewart, C.E., Paustian, K., Conant, R.T., Plante, A.F., Six, J., 2007. Soil carbon saturation: concept, evidence and evaluation. *Biogeochemistry* 86, 19-31.] Do you know the mineral surface area? It would help in interpreting and expanding this work to bioavailability once the sediments are melted if we had some estimates of this. Usually around 50 g C kg⁻¹ clay or silt is near the maximum, but this can vary considerably (as low as 10 and as high as 100 g C kg⁻¹ maybe) based on mineral composition (again pointing to the need to measure this). Examining the values in Table S2, the MAOM concentration ranges from 19.5 and 58.6 g C kg⁻¹ fraction. To me, these are key numbers to base interpretation from. If saturation is near 20 g C kg⁻¹, then we can expect the samples above that to lose MAOM during melt, whereas if 58 g C kg⁻¹ represents saturation, then the 20 g C kg⁻¹ sample could likely take up additional C as MAOM. Again, it could also be variable based on the mineral composition...

4. There is a lot of focus in the paper on the fraction of total OM in the MAOM pool, but this to me is less critical than the total abundance in that pool. Reading the data that way, one might even come to completely opposite conclusions. The authors need to address this and make the case for why they focus on fraction in the MAOM, rather than total abundance.

5. The authors need to clarify why the 1.2m MIS 3 sample does not have characterization data for the MAOM and related to this why this sample, which has the highest abundance of MAOM is often ignored in some of the discussion (at least in my reading). The relationship between the different depths of the samples and the MIS 3 vs. MIS 2/1 are not explained well enough. Should we focus on the 1.2m or 4.6 m samples as key representatives of the core sections? Does the 1.2m MIS 3 represent something close to the 4.6m MIS 2/1?

6. Most of the key findings from the paper derive from the separation of the C into a MAOM pool. Yet, the methodology of this is not mentioned at all in the abstract and the reasons for choosing 6 um as a cutoff vs. density separations vs. a different size cutoff are not discussed. This is key for understanding all the other analyses and the resulting interpretations.

In summary in the general comment area, I think that this paper has some large revisions to be made to bring it to a place where it will support the community of scientists working on this question. First, statistics, or at a very minimum some representation of error on the measurements, is critical even to

properly evaluate the data. Second, the authors need to address the main points above.

LINE COMMENTS:

Abstract: This is well written, but it is not clear how the key component of MAOM was characterized? Can 13-C NMR and 14C analysis delineate MAOM? Would not some other method of fractionation be required (density separation, size separation, etc.) or perhaps some other method to confirm association with minerals? These important details should be included in the abstract.

Ln 26: how is 'strong OM degradation' inferred here?

Ln 28: how is 'less decomposed OM' determined? Presumably via inference of 13-C NMR functional group chemistry, but if so this should again be clarified in the abstract.

Ln 28: Again, what is the definition of MAOM used in this paper? This is critical as there are many methods to do this and a lot of interpretation is required.

Ln 28: On 2nd readthrough, I am trying to see the 'less decomposed' aspect of the late MIS 3. If I look at Fig. 2, C/N ratios of the MAOM are slightly higher for the two samples in the late MIS 3, but is this a significant difference from the early MIS 3? Hard to tell without an errors on the data. The CPI data for the bulk sediments (fig S1) mirror this same trend, although again, what is the error on the measurements? If it is +/- 15% on each (usually a good guess), then some of the values would overlap, suggesting they might not be significantly different.

Ln 28: Next is the statement about there being less MAOM in the late MIS 3 sediments. Looking at Fig. 2, the peak in the abundance of MAOM is actually in the late MIS 3 sediments (1.2m sample). Should I ignore this and focus on the 4.6m sample, then actually it still has plenty of MAOM, more than one of the early MIS 3 samples. In terms of fraction of total C, this is likely the case, but it is not the case because of lower MAOM. That should be made clear as it points to a different mechanism.

Ln 30: 'to association with minerals' instead of 'to MAOM'

Ln 30: Really it is the predicted future gas release that is reduced, not the actual gas release as nothing is changing right?

Main:

Ln 38: 'which has already...'

Ln 40: 'degradation that could increase emissions...'

Ln 50: 'to contain high...'

Ln 57: I think you are actually not testing this. You are exempling the partitioning into particle size fractions. This is a different, although the two are related.

Ln 74: 'when the climate...'

Ln 98: 'MIS 3, the climate...'

Ln 110: 'particles, that likely stabilize OM as MAOM...'

Ln 115: 'oPOM is likely less...' ◊ Otherwise you are implying that the oPOM in the samples you are discussing is less available when that has not been tested.

Ln 120: 'during the Yedoma...'

Ln 145- 145: This is an important point and certainly it deserves more attention throughout. Except that the abundance looks to actually peak at the end of the MIS 3—highest OC in the MAOM pool. Are you disregarding this datapoint? I am confused.

Ln 147-150: OK. Here I am missing on this point and it is a central argument of the paper. If the 1.2 m MIS 2/1 sample is considered—the sample with the greatest fraction of C in MAOM—in the context of representing what happens to permafrost deposits when they are exposed to warm conditions (i.e., what we could expect the MIS 3 samples to look like if they thawed during the present day). Then we could expect them to be transformed into sediments that look like the MIS 2/1 samples. As suggested by the authors, this would drive greater sequestration of C as MAOM, however examining Fig. 2, we are actually going to lose MAOM. From Table S2 we see that the 1.2 m MIS 2/1 sample has ~20 mg OC g⁻¹, which is lower than the average of the MIS 3 samples, and a lot lower than the 1.2 m MIS 3

sample. So assuming that these sediments have the same mineral composition—again we don't know this as the authors have not done any mineral analysis, but what else can be assume—we surmise that either the 1.2 m MIS 2/1 sample is under saturation or that it is actually close to saturation and the other samples are above saturation. In either case, we would expect a loss of MAOM moving from conditions where MIS 3 permafrost melts. This is a very different scenario than presented by the authors, but completely valid. It should be addressed in some way in the text of the paper.

Ln 155: Here I think it would help a lot to calculate the absolute abundance of each functional group by multiplying the rel. ab. by the concentration of OC in that pool. This would make it possible to make and compare statements like this directly. However, we also need errors on these measurements in order to assess if the differences the authors discuss are meaningful.

Ln 157: change 'little' to 'minor'

Ln 162: Is it correct to interpret this as slow degradation? Perhaps another way to achieve the same result would be low turnover of the MAOM pool, corresponding with low overall biological activity. In active surface soils, we often find the MAOM 14C age to be much closer to the fPOM than in subsoils. This is not generally interpreted as slow degradation of the bulk OM, but rather low turnover of the MAOM pool because of lower overall biological activity and low rates of fresh inputs.

Ln 163: 'In the late MIS 3...' and 'deposits, the ages of...'

Ln 164: change 'an' to 'the'

Ln 165: To me, the absolute amount of MAOM is similar or larger in the later MIS 3, so not really a lower level of stabilization as suggested here.

Ln 167: But, isn't the peak of MAOM OC in the late MIS 3? Are you describing this as part of the MIS 2 to 1 period? Need to be more explicit here. The amount of OC 'sequestered as MAOM is actually much lower in the MIS 2 to 1 period, right?

Ln 172: delete 'tremendously'...especially since the highest appears to be at the end of the MIS 3.

Ln 178: This statement should be expanded to include all of the Yedoma deposits as the abundance (not fraction of total C) of MAOM is similar across much of the deposit, with the lowest abundance actually occurring in the MIS 2/1 1.2 m depth!

Final paragraph: The key thing that I am grappling with here is that there is no evidence of lower bioavailability through direct incubations or other means. The lower bioavailability is inferred from association with minerals. This is certainly a good metric---but why not simply expose this material to a 30 day incubation and find out if it is more bioavailable?

Methods: These are informative and sufficient. Good work here.

Fig. 4: In order to interpret this graph and data we need to understand the error on the compound class relative abundance. Also, critically, is not the absolute amount important (concentration X rel. abundance)? Also, why is there no MAOM data for the 1.2 m MIS 3 sample? This is a critical sample that has a high abundance (absolute amount) of MAOM.

Reviewer #3 (Remarks to the Author):

I began this article with some interest because it is a novel idea to begin examining mineral protection mechanisms of permafrost carbon. Mineral protection is a critical factor controlling C turnover in temperate soils but has not been applied to permafrost soils until I saw this paper. The techniques used in this paper are also interesting and appropriate - bulk radiocarbon ages, compound specific radiocarbon analysis, soil separations - these are all very appropriate metrics. The writing style of the paper was only fair, as it was difficult for me, a non-geologist, to understand the different ages and climates. Further, the writing style quickly assumes that all of this historic information about

decomposition and changes in soil properties over geologic time is well known and certain - and it may be - but more context or level of uncertainty of that understanding needs to be provided.

In the end however the critical error I see is the lack of replication, and once I saw this I was not able to take many of the results seriously. If there is no replication (only one core taken in two locations) then I have no faith that any properties between these two sites are different. It could simply be the result of where they happened to place their core barrel that day. It is unfortunate because a lot of work was put into this and I was very excited to really delve into what was learned.

Other comments:

Please try to be more consistent with naming what you are comparing. You use core, location, geologic history, landform, all as synonyms for the two cores. This became confusing.

Please further define the marine isotope stage 1,2,3 and give more context for non-geologists. Is this the best naming convention for these cores?

Line 60: what is a 'high' age?

Figure 5 error bars: obviously this is error due to analytical replication but one cannot compare these two sites without some field replication. True for figures 2-4 as well which lack error bars.

The protection of OM from microbial decay in different fractions should be assessed, even though it has been shown in other studies. How long does that protected carbon stay protected? It's not zero. How much less access to microbes have to OC in OPOM, or any other fraction? And how would accessibility change under real field conditions such as thermokarst processes?

Author response to reviews and resulting edits of *Nature Communications* manuscript “Stabilization of mineral-associated organic carbon in Pleistocene permafrost”

Ref: ms. no. NCOMMS-20-30251-T

Jannik Martens, Carsten W. Mueller, Prachi Joshi, Christoph Rosinger, Markus Maisch, Andreas Kappler, Michael Bonkowski, Georg Schwamborn, Lutz Schirrmeister and Janet Rethemeyer

We gratefully thank the reviewers and the editor for their constructive comments to our initial manuscript submission. We are pleased by the reviewers’ recognition of the overall importance of this study, with comments such as “*I think the findings in this manuscript are interesting and impactful*” (Reviewer 1) or “*clearly this data is needed and will help move the work on permafrost C release upon melting ahead*” (Reviewer 2), and we are grateful for the feedback provided that clearly helped to strengthen this work during revision. Also, reviewer 3 recognizes the importance of this work and finds that “*the techniques used in this paper are interesting and appropriate [...]*” (Reviewer 3), although she/he seems overall concerned about the lack of replication.

Encouraged by the overall constructive feedback, we have followed most of the reviewer suggestions to address the weaknesses of our manuscript through significant additional work, which not only added confidence to our study but also deepened our understanding of organic matter (OM) stabilization in Pleistocene-age permafrost deposits. To address the major concern of the reviewers, **the lack of bioavailability measurements of the mineral-associated OM (MAOM)**, we followed the reviewers’ suggestions and carried out respiration experiments of these fractions. Albeit technically difficult (sodium polytungstate used for density separation eradicates all microbes), we re-vitalize the MAOM by inoculating the fraction using authentic and untreated permafrost material and measured the basal respiration of the MAOM after exhaustion of the inoculum. The results of this experiment clearly add more robust information on the stabilization and bioavailability of MAOM in Pleistocene permafrost.

Moreover, the reviewers indicate that **differences in the mineral composition of the samples may affect MAOM stabilization**. To address this point, we carried out X-ray diffraction analysis of the <6.3 µm fraction to study changes in the mineral composition in the different sediment deposits and, in addition, Mössbauer spectroscopy analysis to distinguish between different species of reactive iron (Fe). The new results suggest that the presence of poorly-crystalline Fe-(oxy)hydrates promoted the stabilization of OC in MAOM during sediment deposition. Our results also indicate that the formation of these minerals and the concurrent stabilization of OC in MAOM varied with changing climatic conditions during the late Pleistocene, which provides a novel conceptual understanding about the stability of OC in permafrost deposits.

Based on our old and new findings, and guided by the reviewer feedback, we have now comprehensively revised our manuscript. All reviewer comments and our responses are listed below, organized such that each reviewer comment is shown first in *italics black font*, followed by our detailed response in normal blue tab-indented text. Our response refers to line numbers in the revised manuscript version.

Reviewer #1:

This manuscript aims to determine the relative importance of mineral associated organic matter in yedoma deposits by analyzing two sediment cores from Siberia, one dated to Marine Isotope Stage 3 (cold & glacial) and one dated to the MIS 2 to 1 transition (warm). The authors analyzed bulk OC, 14C ages (bulk and compound specific) and a suite of n-alkanes on bulk sediment and density/size fractionated sediment to compare OM pools in their two cores.

Yedoma carbon is a Pleistocene-era permafrost OC store that is hypothesized to be relatively labile and a potential source of large amounts of greenhouse gases as Arctic permafrost warms and thaws. The guiding hypothesis for this work is that if a significant amount of this yedoma carbon is mineral-

bound, it will actually be less labile than assumed from previous chemical analyses and therefore potentially more stable and less of a source of greenhouse gases in a warming climate.

In general, the authors conclude that this is true: that much of the yedoma carbon, especially the younger yedoma (from MIS 2-1 transition) is mineral bound and therefore less bioavailable than previously estimated. However, some of the estimates that they cite for the bioavailability of yedoma carbon are not based only chemical studies but instead on microcosm studies, where the kinetics of degradation upon thaw were directly measured. While it is true that mineral bound OM is in general more protected from degradation than free OM, we have direct measurements of the degradation potential of yedoma carbon. It is also true that these direct measurements are from samples that are different from the ones analyzed here, and potentially not directly comparable, but this point is not addressed in the manuscript.

We thank reviewer 1 for this feedback. We understand this comment to criticize that we did not discuss the bioavailability of our sample material at depth, or in the light of earlier incubation experiments based on this type of permafrost sediment. The reviewer is right that the samples used in our study are different from (most) previous studies, which makes it difficult to compare our results to previous incubation experiments, or to discuss the implications of our work in the light of the bioavailability of Yedoma deposits as a whole. However, the main focus of this study is not to discuss the bioavailability of Yedoma, but to study the mechanisms that contribute to OM stability and drive the bioavailability of OM stored in Yedoma. The importance of these stability mechanisms cannot be resolved by direct measurements of the degradation potential. Our work thereby addresses an important knowledge gap and may help to explain why reported CO₂ production varies so greatly between different Yedoma sites and deposits, which is also briefly discussed in the manuscript including the citations of prior incubation work (lines 61-65).

Nonetheless, we recognized that it would be helpful to directly measure the effect of MAOM stabilization to the bioavailability of the OC. Further, we acknowledge that we may have been inconsistent with the use of the word *bioavailability*, which may have caused some confusion about the focus of our study. Thus, we decided to carry out additional work to investigate the bioavailability of OC stored in MAOM using basal respiration measurements of the MAOM, with the results now presented and discussed in the thoroughly revised manuscript (e.g., in Figure 2, Supplementary Table 7, lines 156-160 or 173-175). We have also acted on the point of the use of the word *bioavailability*, and revised our manuscript accordingly (e.g., in line 105, 183 or 199).

Further, the authors give relatively short descriptions with only a shallow treatment of their geochemical data. How does this chemical analysis compare directly with other analyses of the OM chemistry of yedoma carbon? What does this tell us about the chemistries of this OM that is spread out geographically and was deposited over large amounts of time? While the authors do a good job of comparing the OM chemistry of their three time points (early MIS 3, late MIS 3, and MIS2-1 transition) there is very little discussion of how this compares with earlier yedoma carbon studies.

In general, I while I think the findings in this manuscript are interesting and impactful, I think this study would benefit from a longer-form manuscript, with a fuller treatment of the data therein.

We thank the reviewer for this assessment and agree that a deeper discussion about previous work on the topic of Yedoma carbon would be desirable. However, most published work on Yedoma carbon centers around elemental carbon parameters at the bulk level (e.g., Zimov et al., 2006; Schirrmeister et al., 2011a; Kuhry et al., 2013; Strauss et al., 2013), while there is some more recent work on molecular analysis (e.g., Strauss et al., 2015; Stapel et al., 2016, 2018), where such biomarker-based assessments are at best semi-quantitative and cannot provide any insight into OM stability by the different stability mechanism such as MAOM. To the present day, there is not a single study that investigates OC composition and stabilization of different sediment fractions in these permafrost deposits and it is thus not possible to compare with previous work to study stabilization mechanisms. However, we have now

revised our manuscript to include a discussion about the geographical distribution and the implications of this study for the larger Yedoma region; e.g.:

“Our work suggests that OC sequestration as MAOM_{<6.3µm} was inherently involved in the accumulation of OC in permafrost sediments throughout different climates during the late Pleistocene. We therefore hypothesize that MAOM formation in Yedoma occurred concurrently at other sites across Northeastern Siberia and Alaska at the time, likely representing a significant OC pool in the Arctic Yedoma region. Based on published OC stock estimates [...]” (line 253-257)

Further, our revised discussion now acknowledges specifically earlier findings on OM fractions in Yedoma:

“This range agrees well with data published for several permafrost soils (including the seasonally-thawed active layer) from around the circum-Arctic^{33,34,39} and other Yedoma deposits from Northeastern Siberia²⁷, suggesting that MAOM_{<6.3µm} holds a large portion of the OC stored in Yedoma permafrost systems.” (line 143-146).

Reviewer #2:

Review of Martens et al

General comments:

This manuscript presents some good new data on the MAOM fraction of the Yedoma sediments and also provides some organic C characterization of those fractions. Aside from not presenting errors on most of the measurements—something that is desperately needed to understand the relationships between the samples and also a key requirement of NCOMM if I remember correctly—the work appears to be carried out in a highly skilled and careful, rigorous manner. Clearly this data is needed and will help move the work on permafrost C release upon melting ahead. However, as currently presented, this work suffers from some pretty serious interpretive missteps. I have highlighted many of these in the line comments below, and I summarize them briefly here at the top.

1. The authors are not actually measuring the bioavailability of the MAOM. This is inferred based on the size fraction of the OC, and somewhat by the observation that it appears to be already rather degraded based on C/N, 13C NMR, and other C analyses. However, those measures apply to conditions of formation, not the conditions of melting permafrost. To address this the authors look at the MIS 2/1 section of the sample and suggest at one point that this might be what post-thawed Yedoma deposits would look like. As I outline below, this exercise, to me, makes some interpretive errors by focusing on the fraction of total C as MAOM and the net abundance (concentration) of MAOM. I think the latter is the more relevant way to assess the data, and should at a minimum be discussed even if the authors disagree with me. But see below on this and back to the bioavailability point. I certainly agree that MAOM is likely to be of lower bioavailability, but the main issue here is that is not what is measured and the authors are misleading the reader by discussing bioavailability, especially in the abstract and concluding paragraph. Simply discuss this as the amount of C in the fine fraction, which the authors then define as MAOM. Be precise about what is being reported. If the authors had actually measured the bioavailability using an incubation or something, this would really strengthen the paper. If that is possible, I encourage them to do so.

We appreciate the well-informed and very constructive comments provided by reviewer 2, which greatly guided our additional work and the revisions of our paper. Based on the reviewer’s input we have carefully re-assessed our interpretations about the composition and the degradability of the MAOM and we recognize that the molecular and isotopic findings tools are indeed insufficient to

describe the bioavailability of the MAOM, thus limiting the conclusiveness of our previous dataset. Unfortunately, measuring the bioavailability of MAOM using incubations is challenging because the fractionation procedure includes toxic chemicals (sodium polytungstate) that kill the microbial community. Moreover, the representativeness of resulting CO₂ emissions of any OM fraction regarded in isolation rather than in its natural sediment matrix is unclear. After careful consideration of all options, we find that microbial respiration measurements amongst the six investigated sampling depths are the only useful application to reflect how the bioavailability of the MAOM differs between permafrost deposits. We therefore re-activated the 'dead' sediment fractions by adding a slurry solution of untreated, frozen permafrost material. The inoculated fractions were incubated over 108 hours in a climate-controlled respiration chamber, while resulting CO₂ emissions were monitored (now described in line 106-110). This basal respiration was used as a measure for the bioavailability of the MAOM in the permafrost samples, e.g., as quoted below:

“Low inputs of plant litter and strong association of OM with mineral particles is mirrored by relatively low CO₂ production rates during the basal respiration measurements (53±8 and 54±7 μg CO₂-C per gOC per h; Figure 2, Supplementary Table 7).” (line 156-159)

“The very small ¹⁴C age offsets between MAOM_{<6.3μm} and fPOM (0.2 to 1.7 ka; Figure 3) indicate low stabilization of OC during the 48-33 ka period, which is also reflected by higher CO₂ production during incubations of MAOM_{<6.3μm} [...]” (line 172-175)

2. *We do not know the mineral composition of the minerals in the MAOM. Does this matter? Well, if they are strongly Fe-based minerals or comprised of layered silicate clays bound by Fe minerals, then they may be susceptible to release if the material is subjected to Fe reduction in an anoxic melt water, which is a common characteristic of these systems. Further this matters if the mineral composition changes across the different deposits as that may alter the amount of C that can be stored on mineral surfaces. Finally, it matters if the authors are interested in understanding how much of the MAOM will turnover should it become melted as the mineral composition becomes critical in this regard.*

We also considered this important point, which we agree was insufficiently addressed in the previous manuscript version. The reviewer is right that the mineral composition and specifically the availability of reactive iron (Fe) may have significant impact to the formation of MAOM and its bioavailability, although these processes are still under-studied in Arctic environments and permafrost soils. To further investigate the influence of a potentially variable mineral composition and of reactive Fe across different deposits we pursued X-ray diffraction spectroscopy to characterize the whole mineral composition and ⁵⁷Fe Moessbauer spectroscopy in order to determine the different Fe mineral phases. The results show a surprisingly homogenous mineralogy across the depositional time frame from 55,000 – 11,000 years before present (Supplementary Figure 1), and we find an important proportion of poorly crystalline Fe (oxyhydr)oxides across the sample material. These minerals are known to stabilize OC via MAOM (e.g., Chen and Thompson, 2021), although this may change with a shift to anoxic conditions as rightfully pointed out by the reviewer and as recently demonstrated by Patzner et al. (2020) being of relevance for permafrost environments. To i) ground truth the formation of MAOM through Fe minerals in these Pleistocene permafrost deposits, and ii) to discuss changing MAOM stability (and bioavailability) under more anaerobic conditions we have now expanded the discussion to cover these aspects.

For example, the introduction now elaborates on the importance of Fe minerals as part of the MAOM: *“The stabilization of OC as fine-grained (<6.3μm) MAOM is promoted by sorption to reactive iron Fe(III) minerals (e.g., ferrihydrite or goethite)^{36,40,41} and co-precipitation of Fe-OM^{42,43}. However, thawing of permafrost also changes redox conditions due to waterlogged conditions, which may cause Fe(III) mineral reduction and dissolution causing the release of Fe-bound OC⁴⁴⁻⁴⁷.”* (line 81-84).

Further, Fe minerals are now part of the discussion throughout the paper, e.g. in line 154-156: *“The prevalent environmental conditions also resulted in the largest amounts of poorly crystalline Fe(III)*

(oxyhydr)oxides (based on Mössbauer spectra collected at 77K and 5K) in MAOM_{<6.3µm} (35 and 37%) that enhance OM stabilization^{36,40} (Supplementary Table 6). ”.

Finally, the possible dissolution of Fe minerals under potentially anoxic conditions is discussed: “*The wetter, microoxic or even anoxic conditions may be an additional factor leading to lower OC stabilization and higher bioavailability of OM due to reductive dissolution of Fe-OM structures^{44,61}, particularly in thermokarst basins^{18,49}, as shown by a slightly lower content of Fe(III) (oxyhydr)oxides (22 and 28%; measured by Mössbauer spectroscopy at 77K and 5K) and less overall Fe(III) minerals (69 and 70% measured at 77K in 15-11 ka BP vs. 73 and 76% in 55-33 ka BP Yedoma).*” (line 198-203)

3. *The authors have not discussed or assessed the if the C in the fine fraction near its approximate MAOM saturation level? [see e.g., Stewart, C.E., Paustian, K., Conant, R.T., Plante, A.F., Six, J., 2007. Soil carbon saturation: concept, evidence and evaluation. Biogeochemistry 86, 19-31.] Do you know the mineral surface area? It would help in interpreting and expanding this work to bioavailability once the sediments are melted if we had some estimates of this. Usually around 50 g C kg⁻¹ clay or silt is near the maximum, but this can vary considerably (as low as 10 and as high as 100 g C kg⁻¹ maybe) based on mineral composition (again pointing to the need to measure this). Examining the values in Table S2, the MAOM concentration ranges from 19.5 and 58.6 g C kg⁻¹ fraction. To me, these are key numbers to base interpretation from. If saturation is near 20 g C kg⁻¹, then we can expect the samples above that to lose MAOM during melt, whereas if 58 g C kg⁻¹ represents saturation, then the 20 g C kg⁻¹ sample could likely take up additional C as MAOM. Again, it could also be variable based on the mineral composition...*

We largely agree with the reviewer that the MAOM saturation level may be variable between different permafrost samples. As discussed under the previous comment (and as rightfully pointed out by the reviewer here and before), the capacity of MAOM for OC sequestration strongly depends on the availability of the reactive Fe minerals and we therefore decided to focus on Fe biogeochemistry while we have not analyzed the mineral surface area of the fine silt and clay fraction. However, we do know the ratio of OC to elemental Fe, which has thus been used by an increasing number of recent studies to study the sorption capacity of Fe-MAOM (Chen et al., 2020; Patzner et al., 2020; Chen and Thompson, 2021). It is thought that an OC/Fe ratio of 0.22 represents the absorption saturation of MAOM (Kaiser and Guggenberger, 2007; Wagai and Mayer, 2007), while higher numbers may relate to co-precipitation of OM, with contrasting stability/bioavailability under changing redox conditions (Chen et al., 2020). We have now added a brief discussion about the MAOM absorption capacity based on OC/Fe in the manuscript as described below:

“Moreover, the highest OC/Fe mass ratio of 1.1 at 33 ka BP – which is notably higher than the sorption capacity of Fe(III) oxides (0.22)^{59,60} – may show that co-precipitation of Fe-MAOM, in addition to mineral absorption, was more important under such varying oxic conditions⁶¹ than during the previous colder and dryer period.” (line 179-182)

4. *There is a lot of focus in the paper on the fraction of total OM in the MAOM pool, but this to me is less critical than the total abundance in that pool. Reading the data that way, one might even come to completely opposite conclusions. The authors need to address this and make the case for why they focus on fraction in the MAOM, rather than total abundance.*

We agree that the total abundance of the MAOM pool is an important measure to consider. We have previously focused on the relative abundance of the MAOM to make its proportion more comparable among the samples, while the final part of the paper provided an estimated total stock of OC in MAOM in Pleistocene permafrost. However, we feel that both aspects (relative and total abundance) of the MAOM pool should be presented and discussed throughout the paper and we have now revised the discussion accordingly; e.g. in line 169-172: “*This resulted in notably larger amounts of OC in the form of fPOM and oPOM_{<6.3µm} (Figure 2) and in similar, to larger amounts of OC in MAOM_{<6.3µm}, when*

compared to the preceding colder period (55-48 ka) BP, with 8.2 kg m⁻³ (36% of the bulk OC) at 48 ka BP and 14 kg m⁻³ (33%) at 33 ka BP.”

In addition, our revised discussion about MAOM bioavailability and its potential for CO₂ production now includes: “*Despite lower stabilization and bioavailability of the MAOM_{<6.3µm}, we note that the total CO₂-C production potential in the 48-33 ka old sediments is 36% larger (0.55±0.05 and 0.57±0.11 mg CO₂-C per m³ per h) as more MAOM_{<6.3µm} is available per m³ when compared to the cooler period (0.33±0.05 and 0.50±0.06 mg CO₂-C per m³ per h; 55-48 ka BP), which outweighs the differences among the MAOM_{<6.3µm} fractions.” (line 185-189)*

Please also note that Figure 2 provides both pieces of information (the relative and the absolute amount of OC in MAOM), which is now also more clearly mentioned in the figure caption.

5. The authors need to clarify why the 1.2m MIS 3 sample does not have characterization data for the MAOM and related to this why this sample, which has the highest abundance of MAOM is often ignored in some of the discussion (at least in my reading). The relationship between the different depths of the samples and the MIS 3 vs. MIS 2/1 are not explained well enough. Should we focus on the 1.2m or 4.6 m samples as key representatives of the core sections? Does the 1.2m MIS 3 represent something close to the 4.6m MIS 2/1?

With a large proportion of OC in fPOM, sample at 1.2 m of L14-02 is indeed different from most other samples, while the proportion of MAOM and oPOM are more comparable to most other samples. As described above we have now revised our paper to balance between the relative and absolute amount of MAOM throughout the study and we now mention this specifically for the sample at 1.2 m repeatedly throughout the manuscript (e.g., in lines 172, 176 or 180). Regarding the question about which sampling depth is to be considered for each interval, we first like to mention that the different core depths of the two sediment cores represent very different depositional periods where similar depths (below the surface) cannot be correlated between the cores as they originate from different depositional periods in the past. To make the age-depth relationship clearer to the reader, we have removed the Marine Isotope Stage (MIS) based classification and now refer to depositional ages in 1,000 years before present (ka BP) throughout the manuscript, which is also more comprehensible to the broad readership of *Nature Communications*. Further, we have revised Fig. 2 such that these time horizons are now clearly shown.

6. Most of the key findings from the paper derive from the separation of the C into a MAOM pool. Yet, the methodology of this is not mentioned at all in the abstract and the reasons for choosing 6 µm as a cutoff vs. density separations vs. a different size cutoff are not discussed. This is key for understanding all the other analyses and the resulting interpretations.

We agree that it should be made clear to the reader that our work is based on the <6.3 µm fraction upfront in the paper. The 6.3 µm cutoff follows the distinction between coarse and fine silt where our <6.3 µm fraction includes fine silt and clay particles. This follows previous research and work on this topic and applies established laboratory protocols (e.g., von Lützow et al., 2007; Mueller and Koegel-Knabner, 2009; Mueller et al., 2014) that were developed to study MAOM in isolation. Further, this classification makes our results compatible with prior research on this topic. Given the shortness of the abstract (as required by the editorial guidelines) we have now included a very short note about the size (and density) separation applied in this study (line 30 and 33-34). The fractionation is explained at greater depth in the introduction (lines line 78 and moreover in 100), while we refer to the methods for the full description of the fractionation (line 286-302).

In summary in the general comment area, I think that this paper has some large revisions to be made to bring it to a place where it will support the community of scientists working on this question. First, statistics, or at a very minimum some representation of error on the measurements, is critical even to properly evaluate the data. Second, the authors need to address the main points above.

Thank you again for your thorough and well-informed comments, which are now addressed in our revised manuscript submission. Regarding the error representation we have now added more information about the uncertainty of the methods involved, which is explained in further detail below.

LINE COMMENTS:

Abstract: This is well written, but it is not clear how the key component of MAOM was characterized? Can 13-C NMR and 14C analysis delineate MAOM? Would not some other method of fractionation be required (density separation, size separation, etc.) or perhaps some other method to confirm association with minerals? These important details should be included in the abstract.

Thank you for this comment and we agree that the abstract should include this important information. This study characterizes the MAOM using a comprehensive set of methods (incl. NMR, OC and N analysis) to characterize its composition, as well as its stability and bioavailability (¹⁴C, Mössbauer and X-ray spectroscopy, basal respiration measurements), based on size and density separated OM fractions. A (very) short description of this is now included in the revised abstract: “we studied different OM fractions in Siberian permafrost deposits from colder and warmer intervals over the past 55,000 years using organic carbon (OC), nitrogen, ¹⁴C and ¹³C-NMR analysis” (line 30-31) and further “Preservation of OC in the MAOM_{<6.3μm} fraction was enhanced by the presence of reactive iron minerals during cold, dry periods reflected by Fe mineral analysis and low microbial CO₂ production in incubation experiments.” (line 34-36).

Ln 26: how is ‘strong OM degradation’ inferred here?

Ln 28: how is ‘less decomposed OM’ determined? Presumably via inference of 13-C NMR functional group chemistry, but if so this should again be clarified in the abstract.

These points referred to the higher degradation status of the bulk OM based during the 15-11 ka BP interval when compared with the 55-33 ka BP interval. These lines were removed due to major rewriting of the manuscript.

Ln 28: Again, what is the definition of MAOM used in this paper? This is critical as there are many methods to do this and a lot of interpretation is required.

We agree that MAOM should be clearly defined early on in the paper. The revised version now includes a clear definition “mineral-associated in <6.3 μm particles (MAOM_{<6.3μm})” in the abstract (line 33-34), while a more detailed definition is provided in the introduction.

Ln 28: On 2nd readthrough, I am trying to see the ‘less decomposed’ aspect of the late MIS 3. If I look at Fig. 2, C/N ratios of the MAOM are slightly higher for the two samples in the late MIS 3, but is this a significant difference from the early MIS 3? Hard to tell without an errors on the data. The CPI data for the bulk sediments (fig S1) mirror this same trend, although again, what is the error on the measurements? If it is +/- 15% on each (usually a good guess), then some of the values would overlap, suggesting they might not be significantly different.

As described above, the revised abstract includes less discussion about the degradation status of the OM but now focuses more strongly on the stability and bioavailability of the MAOM.

Regarding the error appropriation (as single standard deviation), the uncertainty of the OC measurements in our laboratory were 1% while for N and the combined C/N ratio the uncertainty is 3%, both based on repeatedly run standard materials of three different OC and N concentrations. This information is now added to the manuscript (line 307-309). For the CPI of *n*-alkanes, the uncertainty of the relative abundance of different chain lengths of *n*-alkanes (which is needed for the CPI) is 3%, based on a *n*-alkane standard that was repeatedly run next to our samples (added to line 348-349). Hence, the differences between these samples are statistically significant. Further, we like to note that the

uncertainties of the ^{14}C measurements are included in Supplementary Table 3, while the uncertainty of the basal respiration measurements is included in the revised manuscript as well.

Ln 28: Next is the statement about there being less MAOM in the late MIS 3 sediments. Looking at Fig. 2, the peak in the abundance of MAOM is actually in the late MIS 3 sediments (1.2m sample). Should I ignore this and focus on the 4.6m sample, then actually it still has plenty of MAOM, more than one of the early MIS 3 samples. In terms of fraction of total C, this is likely the case, but it is not the case because of lower MAOM. That should be made clear as it points to a different mechanism.

The reviewer is right that the total amount of OC in MAOM is larger in the 1.2 m sample. As described in detail above, the revised manuscript now includes a more balanced discussion about the relative and absolute abundance of the OC pool in MAOM to alleviate such misunderstandings (line 186-189).

Ln 30: 'to association with minerals' instead of 'to MAOM'

This line was removed due to rewriting of the abstract.

Ln 30: Really it is the predicted future gas release that is reduced, not the actual gas release as nothing is changing right?

This line refers to a reduced potential for future greenhouse gas release upon anticipated thaw of these permafrost deposits. However, the conclusive sentence of the abstract was rewritten during manuscript revisions.

Main:

Ln 38: 'which has already...'

The word *has* was added to the line.

Ln 40: 'degradation that could increase emissions...'

This line was revised accordingly.

Ln 50: 'to contain high...'

The line was revised to “*to contain large amounts of labile compounds*”.

Ln 57: I think you are actually not testing this. You are exempling the partitioning into particle size fractions. This is a different, although the two are related.

As described above, our revised manuscript now includes the measurements required to address MAOM.

Ln 74: 'when the climate...'

The word *the* was added accordingly.

Ln 98: 'MIS 3, the climate...'

The word *the* was added accordingly.

Ln 110: 'particles, that likely stabilize OM as MAOM...'

Large parts of the discussion were rewritten, including this line.

Ln 115: *'oPOM is likely less...'* Otherwise you are implying that the oPOM in the samples you are discussing is less available when that has not been tested.

Also this line was removed due to major rewriting of the discussion. However, we no longer refer to the oPOM fractions using the word *bioavailable*, as this has not been tested for this fraction using incubations/respiration measurements for the technical challenges described above.

Ln 120: *'during the Yedoma...'*

The word *the* was added accordingly.

Ln 145- 145: *This is an important point and certainly it deserves more attention throughout. Except that the abundance looks to actually peak at the end of the MIS 3—highest OC in the MAOM pool. Are you disregarding this datapoint? I am confused.*

As described above, the revised manuscript is now more balanced between absolute and relative amounts of OC in the MAOM to the total bulk OC.

Ln 147-150: *OK. Here I am missing on this point and it is a central argument of the paper. If the 1.2 m MIS 2/1 sample is considered—the sample with the greatest fraction of C in MAOM—in the context of representing what happens to permafrost deposits when they are exposed to warm conditions (i.e., what we could expect the MIS 3 samples to look like if they thawed during the present day). Then we could expect them to be transformed into sediments that look like the MIS 2/1 samples. As suggested by the authors, this would drive greater sequestration of C as MAOM, however examining Fig. 2, we are actually going to lose MAOM. From Table S2 we see that the 1.2 m MIS 2/1 sample has ~20 mg OC g⁻¹, which is lower than the average of the MIS 3 samples, and a lot lower than the 1.2 m MIS 3 sample. So assuming that these sediments have the same mineral composition—again we don't know this as the authors have not done any mineral analysis, but what else can be assume—we surmise that either the 1.2 m MIS 2/1 sample is under saturation or that it is actually close to saturation and the other samples are above saturation. In either case, we would expect a loss of MAOM moving from conditions where MIS 3 permafrost melts. This is a very different scenario than presented by the authors, but completely valid. It should be addressed in some way in the text of the paper.*

This is a misunderstanding. The deglacial sediments (15-11 ka BP; MIS 2/1) are by no means a “degraded” version of the 55-33 ka BP (MIS 3) material but represent sediments that formed during the deglacial period. However, the reviewer is right that the total amount of OC in MAOM decreased from the 55-33 ka BP to the 15-11 ka BP, and even within the 15-11 ka BP interval. This is now clearly stated in the opening sentence of the largely re-written paragraph about MAOM in the 15-11 ka BP section:

“During the deglacial period, the amount of OC in MAOM_{<6.3µm} decreased from highest values of 11 kg m⁻³ (74% of total OC) around 15 ka BP to lowest values of 1.9 kg m⁻³ in the youngest sediment investigated from around 11 ka BP (45% of total OC; Figure 2).” (line 192-194)

Moreover, based on the mineral analysis and bioavailability measurements now included in the manuscript we can add that i) the general mineral composition is indeed similar across all samples (based on XRD), while ii) Mössbauer Fe-mineral analysis show differences between the two permafrost cores where the 15-11 ka BP section includes less reactive iron, which leads to iii) higher basal respiration and thus higher bioavailability of the MAOM in these sediments. These new perspectives and conclusions are now included in the largely rewritten paragraph about MAOM in the 15-11 ka BP sediments (lines 192-208).

Ln 155: *Here I think it would help a lot to calculate the absolute abundance of each functional group by multiplying the rel. ab. by the concentration of OC in that pool. This would make it possible to make and compare statements like this directly. However, we also need errors on these measurements in order to assess if the differences the authors discuss are meaningful.*

This particular point is no longer part of the manuscript due to major revision. However, we have revised the manuscript to also consider the absolute concentrations of OC and concentrations of the functional groups whenever differences between samples were unclear (e.g., in line 182-185). As described above, the errors of the OC measurements were small; and this information is now added to the manuscript.

Ln 157: change 'little' to 'minor'

The line was rewritten during the revisions.

Ln 162: Is it correct to interpret this as slow degradation? Perhaps another way to achieve the same result would be low turnover of the MAOM pool, corresponding with low overall biological activity. In active surface soils, we often find the MAOM 14C age to be much closer to the fPOM than in subsoils. This is not generally interpreted as slow degradation of the bulk OM, but rather low turnover of the MAOM pool because of lower overall biological activity and low rates of fresh inputs.

We agree with the understanding of the reviewer of the ¹⁴C data to reflect turnover. The use of the word *slow* is unconventional and misleading in this sense. We have thus revised our choice of words accordingly throughout the manuscript. Further, we now included the statement “*This fine silt and clay sized MAOM_{<6.3μm} consisted of substantially decomposed OM³⁸ with higher ¹⁴C ages due to lower microbial turnover*” (line 78-79) to alleviate any more misunderstandings.

Ln 163: 'In the late MIS 3...' and 'deposits, the ages of...'

The line was rewritten during the revisions.

Ln 164: change 'an' to 'the'

The line was rewritten during the revisions.

Ln 165: To me, the absolute amount of MAOM is similar or larger in the later MIS 3, so not really a lower level of stabilization as suggested here.

Ln 167: But, isn't the peak of MAOM OC in the late MIS 3? Are you describing this as part of the MIS 2 to 1 period? Need to be more explicit here. The amount of OC 'sequestered as MAOM is actually much lower in the MIS 2 to 1 period, right?

As described above, the revised manuscript is now balanced between absolute and relative amounts of OC in the MAOM to the total bulk OC.

Ln 172: delete 'tremendously' ...especially since the highest appears to be at the end of the MIS 3.

The word was deleted during revision.

Ln 178: This statement should be expanded to include all of the Yedoma deposits as the abundance (not fraction of total C) of MAOM is similar across much of the deposit, with the lowest abundance actually occurring in the MIS 2/1 1.2 m depth!

The estimated pool of OC in MAOM of all permafrost deposits are based on the relative amounts found in our study that was multiplied with the published concentration estimates of the total OC in these sediments across Siberia and Alaska. We believe that the revised version of this conclusive paragraph is now more clear on that point:

“Based on published OC stock estimates² and mass contributions of MAOM_{<6.3μm} in this study, we estimate that about 40±21 Gt of the OC that is freeze-locked in Yedoma may be present as MAOM_{<6.3μm} (Figure 6; Supplementary Table 8). By contrast, post-glacial thermokarst sediments may hold 77±44 Gt OC as MAOM_{<6.3μm}.” (line 257-260)

Final paragraph: The key thing that I am grappling with here is that there is no evidence of lower bioavailability through direct incubations or other means. The lower bioavailability is inferred from association with minerals. This is certainly a good metric---but why not simply expose this material to a 30 day incubation and find out if it is more bioavailable?

As described under the overarching comments, we carried out bioavailability measurements based on basal respiration experiments of the MAOM.

Methods: These are informative and sufficient. Good work here.

We appreciate this assessment!

Fig. 4: In order to interpret this graph and data we need to understand the error on the compound class relative abundance. Also, critically, is not the absolute amount important (concentration X rel. abundance)? Also, why is there no MAOM data for the 1.2 m MIS 3 sample? This is a critical sample that has a high abundance (absolute amount) of MAOM.

Figure 4 was entirely removed from the manuscript during manuscript revisions. However, we have redesigned a Figure 2 such to also include the NMR results of all MAOM, shown next to amount of OC in MAOM to allow a comparison of the total abundance. As detailed above, information about the error of the OC measurements is now included in the revised manuscript.

Reviewer #3:

I began this article with some interest because it is a novel idea to begin examining mineral protection mechanisms of permafrost carbon. Mineral protection is a critical factor controlling C turnover in temperate soils but has not been applied to permafrost soils until I saw this paper. The techniques used in this paper are also interesting and appropriate - bulk radiocarbon ages, compound specific radiocarbon analysis, soil separations - these are all very appropriate metrics. The writing style of the paper was only fair, as it was difficult for me, a non-geologist, to understand the different ages and climates. Further, the writing style quickly assumes that all of this historic information about decomposition and changes in soil properties over geologic time is well known and certain - and it may be - but more context or level of uncertainty of that understanding needs to be provided.

In the end however the critical error I see is the lack of replication, and once I saw this I was not able to take many of the results seriously. If there is no replication (only one core taken in two locations) then I have no faith that any properties between these two sites are different. It could simply be the result of where they happened to place their core barrel that day. It is unfortunate because a lot of work was put into this and I was very excited to really delve into what was learned.

We thank the reviewer for providing their perspective. The reviewer raises serious concerns about the replicability of our results, which we believe are rooted in misunderstanding of our and previous work on terrestrial paleo-environmental research. First and foremost, the focus of our work is not about the differences of the properties between the two permafrost drill cores, but about the well-described and well-dated depositional periods that these cores represent. As described throughout the paper, the dating of these sediment cores is based on ¹⁴C analysis of plant fragments (lines 271-283) and is further cross-validated by stratigraphic correlation with a nearby coastal site that was repeatedly studied and described in 10+ studies (e.g., Meyer et al., 2002; Schirrmeister et al., 2002; Ilyashuk et al., 2006; Andreev et al., 2009; Wetterich et al., 2011, 2014; Schwamborn and Wetterich, 2015; Zimmermann et al., 2017; Stapel et al., 2018; Walz et al., 2018). Secondly, we like to stress that the OM characteristics of our sample material are well in line with a large number of biogeochemical studies on OC quantities, its sources and compositional characteristics, as well as its spatial and stratigraphic variability in late

Pleistocene permafrost (Yedoma) deposits. It is also important to note that these permafrost deposits represent a sediment facies characteristic for the continuous permafrost zone in Siberia and North America, for which the OM characteristics and their variability are well-described due to the extensive work carried out by our colleagues with hundreds of spatially-distributed observations (Schirrmeister et al., 2011b, 2011a, 2013; Strauss et al., 2013, 2017). The sample material retrieved from our two permafrost drill cores were intentionally selected to cover the three time scales that are the most important periods of Yedoma sedimentation (Schirrmeister et al., 2011b, 2017; Zimmermann et al., 2017; Stapel et al., 2018). In the context of the extensively-studied stratigraphy of the study site, it is also worth mentioning that the bulk OC data produced in this study (OC concentrations, OC/N ratios and CPI values) are in agreement with published records from this site and align with the broader Yedoma region. To further illustrate the local stratigraphy, we have included a stratigraphic overview of the study site below; a simplified version is included in Figure 1 of the manuscript.

Figure 1: Overview of the local stratigraphy of Bol'shoy Lyakhovsky Island in Northeastern Siberia (from Schwamborn and Wetterich, 2015). Shown are the locations of the two permafrost cores used in this study and locations of previous works that contribute to this stratigraphic framework.

As demonstrated in the writing above and supported by the published literature, there is high confidence that the OM in our sample material from the two permafrost cores is not composed randomly (as it is suggested by the reviewer) but reflects past climatic and environmental conditions.

Finally, we like to add that this is not the first study on OM stabilization mechanisms in permafrost regions. As described and discussed throughout the paper, there are a number of papers (e.g., Höfle et al., 2013; Gentsch et al., 2015a, 2015b, 2018; Mueller et al., 2015) that used very similar techniques as the present study. We find our results to be consistent with these previous findings, which adds further confidence to the findings of this first study on OM stability in deeper permafrost deposits.

After careful scrutiny of our manuscript, we admit that the representativeness of our sampling site for the larger Yedoma region in Northeastern Siberian and Alaska may have been unclear to the reader. To clarify this, we have now revised the introduction accordingly: “Core L14-02 was recovered from a Yedoma hill that comprises undisturbed Yedoma sediments deposited during a cold period 33-55 ka before present (BP) for which widespread sediment and OM deposition is documented across large areas in Northeastern Siberia^{2,17}.” (line 90-93). Also, we have revised parts of the discussion and conclusion to clarify that the stability mechanisms found at our study site were likely of importance also for the larger Yedoma region (line 253-257). Further, a line about previous findings of OC in MAOM at other Yedoma sites in Northeastern Siberia was added to the discussion (line 145), which further supports our findings.

To improve the clarity of the different depositional time frames for non-geology readers, we have revised the manuscript such that the different core depths are now classified based on their depositional time frame (in 1,000 years before present – ka BP) instead of the classification in Marine Isotope Stages (MIS).

Other comments:

Please try to be more consistent with naming what you are comparing. You use core, location, geologic history, landform, all as synonyms for the two cores. This became confusing.

We appreciate this comment that the sample labelling was confusing. As described above, we have now revised our manuscript such that we removed the MIS classification. Moreover, the introduction was expanded as described above.

Please further define the marine isotope stage 1,2,3 and give more context for non-geologists. Is this the best naming convention for these cores?

We agree that the concept of Marine Isotope Stages (MIS) may be difficult to understand, although this concept was introduced in the original manuscript. The revised manuscript no longer follows this classification and instead uses the depositional time frame to describe the age of the sediments.

Line 60: what is a 'high' age?

In the context of OM fractions, a high ^{14}C age represents depletion of ^{14}C due to low turnover of the OC. We acknowledge that this use may cause some confusion for non-geology readers and we have now revised this line to “with higher ^{14}C ages due to lower microbial turnover [...]” (line 79).

Figure 5 error bars: obviously this is error due to analytical replication but one cannot compare these two sites without some field replication. True for figures 2-4 as well which lack error bars.

The error bar in this figure (now Figure 4 in the revised manuscript) is not based on analytical replication but on the variability of OC concentrations based on a large database with hundreds of observations (Strauss et al., 2013). Hence, this figure (and our stock estimates for OC in MAOM it shows) includes reasonable uncertainties for the stock estimates, while all other figures present results for our study site only. We now realize that the error appropriation was insufficiently explained in the caption of Figure 5, which is now explained in the revised version of the caption. For Figure 2, however, it is difficult to represent error bars in stacked bar plots, while this is the most efficient and the conventional way of presenting this kind of data. Instead, we have now included information about the uncertainty of the different measurements to the Methods (e.g., in line 307-309 or 348-349).

The protection of OM from microbial decay in different fractions should be assessed, even though it has been shown in other studies. How long does that protected carbon stay protected? It's not zero. How much less access to microbes have to OC in OPOM, or any other fraction? And how would accessibility change under real field conditions such as thermokarst processes?

The reviewer is right that OC stabilization may also occur in oPOM via particle occlusion, as it is explained throughout the manuscript (e.g., line 68-70), while fPOM is generally seen as a labile fraction without any protection. However, OC in MAOM is clearly the dominating fraction (>50% of the bulk OC) and we decided to focus our work on the stabilization and mechanics of OC in this pool.

The reviewer is also right that the OC stored in MAOM is not entirely protected from microbial degradation. This is, for instance, shown by the respiration measurements we have carried out in response to the comments provided by reviewer 2. Moreover, we find that the environmental conditions at the time of the deposition exert first-order control on the stability of Fe-MAOM, particularly under influence of thermokarst processes. These aspects of OC stabilization and MAOM protection are now addressed throughout the discussion of the revised manuscript (e.g., 156-161, or 172-177). Further, we have re-organized parts of the paper such that the occlusion of particulate OM is now discussed separately (line 211-229).

References

- Andreev, A. A., Grosse, G., Schirrmeister, L., Kuznetsova, T. v., Kuzmina, S. A., Bobrov, A. A., Tarasov, P. E., Novenko, E. Y., Meyer, H., Derevyagin, A. Y., Kienast, F., Bryantseva, A. and Kunitsky, V. v: Weichselian and Holocene palaeoenvironmental history of the Bol'shoy Lyakhovsky Island, New Siberian Archipelago, Arctic Siberia, *Boreas*, 38(1), 72–110 [online] Available from: <http://dx.doi.org/10.1111/j.1502-3885.2008.00039.x>, 2009.
- Chen, C. and Thompson, A.: The influence of native soil organic matter and minerals on ferrous iron oxidation, *Geochim Cosmochim Acta*, 292, 254–270, doi:10.1016/j.gca.2020.10.002, 2021.
- Chen, C., Hall, S. J., Coward, E. and Thompson, A.: Iron-mediated organic matter decomposition in humid soils can counteract protection, *Nat Commun*, 11(1), 1–13, doi:10.1038/s41467-020-16071-5, 2020.
- Dutta, K., Schuur, E. A. G., NEFF, J. C. and Zimov, S. A.: Potential carbon release from permafrost soils of Northeastern Siberia, *Glob Chang Biol*, 12(12), 2336–2351, doi:10.1111/j.1365-2486.2006.01259.x, 2006.
- Gentsch, N., Wild, B., Mikutta, R., Čapek, P., Diáková, K., Schrumpf, M., Turner, S., Minnich, C., Schaarschmidt, F., Shibistova, O., Schnecker, J., Urich, T., Gittel, A., Šantrůčková, H., Bárta, J., Lashchinskiy, N., Fuß, R., Richter, A. and Guggenberger, G.: Temperature response of permafrost soil carbon is attenuated by mineral protection, *Glob Chang Biol*, doi:10.1111/gcb.14316, 2018.
- Gentsch, N., Mikutta, R., Shibistova, O., Wild, B., Schnecker, J., Richter, A., Urich, T., Gittel, A., Šantrůčková, H., Bárta, J., Lashchinskiy, N., Mueller, C. W., Fuß, R. and Guggenberger, G.: Properties and bioavailability of particulate and mineral-associated organic matter in Arctic permafrost soils, Lower Kolyma Region, Russia, *Eur J Soil Sci*, 66(4), 722–734, doi:10.1111/ejss.12269, 2015a.
- Gentsch, N., Mikutta, R., Alves, R. J. E., Barta, J., Čapek, P., Gittel, A., Hugelius, G., Kuhry, P., Lashchinskiy, N., Palmtag, J., Richter, A., Šantrůčková, H., Schnecker, J., Shibistova, O., Urich, T., Wild, B. and Guggenberger, G.: Storage and transformation of organic matter fractions in cryoturbated permafrost soils across the Siberian Arctic, *Biogeosciences*, 12(14), 4525–4542, doi:10.5194/bg-12-4525-2015, 2015b.
- Höfle, S., Rethemeyer, J., Mueller, C. W. and John, S.: Organic matter composition and stabilization in a polygonal tundra soil of the Lena Delta, *Biogeosciences*, 10(5), 3145–3158, doi:10.5194/bg-10-3145-2013, 2013.
- Ilyashuk, B. P., Andreev, A. A., Bobrov, A. A., Tumskoy, V. E. and Ilyashuk, E. A.: Interglacial History of a Palaeo-lake and Regional Environment: A Multi-proxy Study of a Permafrost Deposit from Bol'shoy Lyakhovsky Island, Arctic Siberia, *J Paleolimnol*, 35(4), 855–872 [online] Available from: <http://link.springer.com/10.1007/s10933-005-5859-6>, 2006.
- Kaiser, K. and Guggenberger, G.: Sorptive stabilization of organic matter by microporous goethite: Sorption into small pores vs. surface complexation, *Eur J Soil Sci*, 58(1), 45–59, doi:10.1111/j.1365-2389.2006.00799.x, 2007.
- Kuhry, P., Grosse, G., Harden, J. W., Hugelius, G., Koven, C. D., Ping, C.-L., Schirrmeister, L. and Tarnocai, C.: Characterisation of the Permafrost Carbon Pool, *Permafr Periglac Process*, 24(2), 146–155, doi:10.1002/ppp.1782, 2013.
- von Lützw, M., Kögel Knabner, I., Ekschmitt, K., Flessa, H., Guggenberger, G., Matzner, E. and Marschner, B.: SOM fractionation methods: Relevance to functional pools and to stabilization mechanisms, *Soil Biol Biochem*, 39(9), 2183–2207 [online] Available from: <http://linkinghub.elsevier.com/retrieve/pii/S0038071707001125>, 2007.

- Meyer, H., Dereviagin, A., Siegert, C., Schirrmeister, L. and Hubberten, H. W.: Palaeoclimate reconstruction on Big Lyakhovsky Island, north Siberia—hydrogen and oxygen isotopes in ice wedges, *Permafrost Periglacial Process*, 13(2), 91–105 [online] Available from: <http://doi.wiley.com/10.1002/ppp.416>, 2002.
- Mueller, C. W. and Koegel-Knabner, I.: Soil organic carbon stocks, distribution, and composition affected by historic land use changes on adjacent sites, *Biol Fertil Soils*, 45(4), 347–359 [online] Available from: <http://link.springer.com/10.1007/s00374-008-0336-9>, 2009.
- Mueller, C. W., Rethemeyer, J., Kao-Kniffin, J., Löppmann, S., Hinkel, K. M. and G Bockheim, J.: Large amounts of labile organic carbon in permafrost soils of northern Alaska, *Glob Chang Biol*, 21(7), 2804–2817 [online] Available from: <http://doi.wiley.com/10.1111/gcb.12876>, 2015.
- Mueller, C. W., Gutsch, M., Kothieringer, K., Leifeld, J., Rethemeyer, J., Brueggemann, N. and Kögel Knabner, I.: Bioavailability and isotopic composition of CO₂ released from incubated soil organic matter fractions, *Soil Biol Biochem*, 69, 168–178 [online] Available from: <http://linkinghub.elsevier.com/retrieve/pii/S0038071713004136>, 2014.
- Patzner, M. S., Mueller, C. W., Malusova, M., Baur, M., Nikeleit, V., Scholten, T., Hoeschen, C., Byrne, J. M., Borch, T., Kappler, A. and Bryce, C.: Iron mineral dissolution releases iron and associated organic carbon during permafrost thaw, *Nat Commun*, 11(1), 1–11, doi:10.1038/s41467-020-20102-6, 2020.
- Schirrmeister, L., Grosse, G., Wetterich, S., Overduin, P. P., Strauss, J., Schuur, E. A. G. and Hubberten, H.-W. W.: Fossil organic matter characteristics in permafrost deposits of the northeast Siberian Arctic, *J Geophys Res Biogeosci*, 116(G2), G00M02, doi:10.1029/2011JG001647, 2011a.
- Schirrmeister, L., Oezen, D. and Geyh, M. A.: ²³⁰Th/^U Dating of Frozen Peat, Bol'shoy Lyakhovsky Island (Northern Siberia), *Quat Res*, 57(2), 253–258 [online] Available from: <http://www.sciencedirect.com/science/article/pii/S0033589401923063>, 2002.
- Schirrmeister, L., Kunitsky, V., Grosse, G., Wetterich, S., Meyer, H., Schwamborn, G., Babiy, O., Derevyagin, A. and Siegert, C.: Sedimentary characteristics and origin of the Late Pleistocene Ice Complex on north-east Siberian Arctic coastal lowlands and islands - A review, *Quaternary International*, 241(1–2), 3–25, doi:10.1016/j.quaint.2010.04.004, 2011b.
- Schirrmeister, L., Froese, D., Tumskoy, V., Grosse, G. and Wetterich, S.: Yedoma: Late Pleistocene ice-rich syngenetic permafrost of Beringia, *EPIC3 Encyclopedia of Quaternary Science*. 2nd edition, 542–552, doi:10.1016/B978-0-444-53643-3.00106-0, 2013.
- Schirrmeister, L., Schwamborn, G., Overduin, P. P., Strauss, J., Fuchs, M. C., Grigoriev, M., Yakshina, I., Rethemeyer, J., Dietze, E. and Wetterich, S.: Yedoma Ice Complex of the Buor Khaya Peninsula (southern Laptev Sea), *Biogeosciences*, 14(5), 1261–1283, doi:10.5194/bg-14-1261-2017, 2017.
- Schwamborn, G. and Wetterich, S.: Russian-German cooperation CARBOPERM: field campaigns to Bol'shoy Lyakhovsky Island in 2014, , 686 [online] Available from: <http://epic.awi.de/37311/>, 2015.
- Stapel, J. G., Schirrmeister, L., Overduin, P. P., Wetterich, S., Strauss, J., Horsfield, B. and Mangelsdorf, K.: Microbial lipid signatures and substrate potential of organic matter in permafrost deposits - implications for future greenhouse gas production, *J Geophys Res Biogeosci* [online] Available from: <http://doi.wiley.com/10.1002/2016JG003483>, 2016.
- Stapel, J. G., Schwamborn, G., Schirrmeister, L., Horsfield, B. and Mangelsdorf, K.: Substrate potential of last interglacial to Holocene permafrost organic matter for future microbial greenhouse gas production, *Biogeosciences*, 15(7), 1969–1985, doi:10.5194/bg-15-1969-2018, 2018.

- Strauss, J., Schirrmeister, L., Mangelsdorf, K., Eichhorn, L., Wetterich, S. and Herzsuh, U.: Organic matter quality of deep permafrost carbon – a study from Arctic Siberia, *Biogeosciences*, 12, 2227–2245, doi:10.5194/bg-12-2227-2015, 2015.
- Strauss, J., Schirrmeister, L., Grosse, G., Wetterich, S., Ulrich, M., Herzsuh, U. and Hubberten, H.-W.: The deep permafrost carbon pool of the Yedoma region in Siberia and Alaska, *Geophys Res Lett*, 40(23), 6165–6170, doi:10.1002/2013GL058088, 2013.
- Strauss, J., Schirrmeister, L., Grosse, G., Fortier, D., Hugelius, G., Knoblauch, C., Romanovsky, V., Schädel, C., von Deimling, T. S., Schuur, E. A. G., Shmelev, D., Ulrich, M. and Veremeeva, A.: Deep Yedoma permafrost: A synthesis of depositional characteristics and carbon vulnerability, *Earth Sci Rev*, 172, 75–86, doi:10.1016/j.earscirev.2017.07.007, 2017.
- Wagai, R. and Mayer, L. M.: Sorptive stabilization of organic matter in soils by hydrous iron oxides, *Geochim Cosmochim Acta*, 71(1), 25–35, doi:10.1016/j.gca.2006.08.047, 2007.
- Walz, J., Knoblauch, C., Tigges, R., Opel, T., Schirrmeister, L. and Pfeiffer, E. M.: Greenhouse gas production in degrading ice-rich permafrost deposits in northeastern Siberia, *Biogeosciences*, 15(17), 5423–5436, doi:10.5194/bg-15-5423-2018, 2018.
- Wetterich, S., Tumskey, V., Rudaya, N., Andreev, A. A., Opel, T., Meyer, H., Schirrmeister, L. and Hüls, M.: Ice Complex formation in arctic East Siberia during the MIS3 Interstadial, *Quat Sci Rev*, 84, 39–55 [online] Available from: <http://www.sciencedirect.com/science/article/pii/S0277379113004526>, 2014.
- Wetterich, S., Rudaya, N., Tumskey, V., Andreev, A. A., Opel, T., Schirrmeister, L. and Meyer, H.: Last Glacial Maximum records in permafrost of the East Siberian Arctic, *Quat Sci Rev*, 30(21–22), 3139–3151 [online] Available from: <http://www.sciencedirect.com/science/article/pii/S0277379111002319>, 2011.
- Zimmermann, H., Raschke, E., Epp, L. S., Stoof-Leichsenring, K. R., Schirrmeister, L., Schwamborn, G. and Herzsuh, U.: The history of tree and shrub taxa on bol'shoy lyakhovsky Island (New Siberian Archipelago) since the last interglacial uncovered by sedimentary ancient DNA and pollen data, *Genes (Basel)*, 8(10), 273, doi:10.3390/genes8100273, 2017.
- Zimov, S. A., Davydov, S. P., Zimova, G. M., Davydova, A. I., Schuur, E. A. G., Dutta, K. and Chapin, I. S.: Permafrost carbon: Stock and decomposability of a globally significant carbon pool, *Geophys Res Lett*, 33(20), L20502, doi:10.1029/2006GL027484, 2006.

REVIEWERS' COMMENTS

Reviewer #1 (Remarks to the Author):

The authors have significantly updated and improved this manuscript upon resubmission, undertaking additional experiments in order to improve the discussion of OC bioavailability of mineral-bound organic matter. Further, the results & discussion has been rewritten to include discussion of geographical distribution as well as other yedoma literature.

I found no major issues with the manuscript in its revised form - I think the authors have done a great job of improving and addressing reviewer concerns.

Reviewer #3 (Remarks to the Author):

This article continues to interest me and I appreciate all the work that the authors did to respond to my comments and those of the other reviewers. I am still somewhat concerned about their extrapolation to the entire Arctic from one or two cores (supp table 8 and conclusions), and I would ask that they add some text to include the caveat this information comes from only two cores and improved accuracy will come with additional studies. The patterns from these two cores are however quite interesting and will spur further research. I would also ask that the authors closely check their text for grammatical errors, sometimes related to a misplaced comma or missing word.

Author response to reviews and resulting edits of *Nature Communications* manuscript “Stabilization of mineral-associated organic carbon in Pleistocene permafrost”

Ref: ms. no. NCOMMS-20-30251A-Z

Jannik Martens, Carsten W. Mueller, Prachi Joshi, Christoph Rosinger, Markus Maisch, Andreas Kappler, Michael Bonkowski, Georg Schwamborn, Lutz Schirmer and Janet Rethemeyer

We gratefully thank the editor and the reviewers for re-considering our revised manuscript. We are pleased that reviewer 1 is now fully backing publication, and we are very delighted that also reviewer 3 is now supporting publication after addressing two remaining minor issues. These issues are now addressed in this final manuscript revision, which further improved quality and clarity of our paper.

Reviewer #1:

The authors have significantly updated and improved this manuscript upon resubmission, undertaking additional experiments in order to improve the discussion of OC bioavailability of mineral-bound organic matter. Further, the results & discussion has been rewritten to include discussion of geographical distribution as well as other yedoma literature. I found no major issues with the manuscript in its revised form - I think the authors have done a great job of improving and addressing reviewer concerns.

We gratefully thank reviewer 1 for reading and re-considering our revised work. We are very pleased and feel encouraged by this assessment!

Reviewer #3:

This article continues to interest me and I appreciate all the work that the authors did to respond to my comments and those of the other reviewers. I am still somewhat concerned about their extrapolation to the entire Arctic from one or two cores (supp table 8 and conclusions), and I would ask that they add some text to include the caveat this information comes from only two cores and improved accuracy will come with additional studies. The patterns from these two cores are however quite interesting and will spur further research. I would also ask that the authors closely check their text for grammatical errors, sometimes related to a misplaced comma or missing word.

We are pleased to hear that our efforts and the revised manuscript are well received. We again like to thank the reviewer for the constructive feedback, which greatly improved the clarity around the age-depth intervals and corresponding climate periods in our manuscript. However, we also agree with the reviewer that extrapolating results of our study site to other regions (where similar permafrost deposits are abundant) introduces further uncertainty. We followed the reviewer's advice and revised the discussion/conclusion to improve clarity around the extrapolation and its associated uncertainty to readers:

“Based on published stock estimates of OC in Yedoma sediments² and mass contributions of MAOM_{<6.3μm} at our study site, we estimate that about 40±21 Gt of the OC that is freeze-locked in Yedoma may be present as MAOM_{<6.3μm} (Figure 6; Supplementary Table 8). By contrast, post-glacial thermokarst sediments may hold 77±44 Gt OC as MAOM_{<6.3μm}. We like to emphasize that these numbers build on a limited observational dataset and constitute first-order estimates that are attributed to uncertainty. While future research will improve accuracy of such estimates, the present study suggests that a total of about 117±65 Gt OC in ice-rich permafrost deposits may be stored as MAOM, which adds complexity to the bioavailability of the sequestered OC. Contrasting stability and bioavailability of Pleistocene-age OC deposited during different climate periods should be considered when anticipating future permafrost thaw and the potential for greenhouse gas production and climate-carbon feedback.” (lines 256-266).

Finally, we thank the reviewer for pointing out grammatical errors. We have paid extra attention to any remaining grammatical issues and correct those throughout the manuscript in this final revision.